# Reinforcement Learning of Industrial Sequential Decision-making tasks under Near-predictable Dynamics: a Bi-Critic variance reduction Approach

## Abstract

Learning to plan and schedule is receiving increasing attention for industrial decision-making tasks (partly) for its potential to outperform heuristics, especially under dynamic uncertainty, as well as its efficiency in problem-solving, particularly with the adoption of neural networks and the behind GPU computing. Naturally, reinforcement learning (RL) with the Markov decision process (MDP) becomes a popular paradigm. Instead of handling the near-stationary environments like Atari games or the opposite case for open world dynamics with high uncertainty, in this paper, we aim to devise a tailored RL-based approach for the practice setting in the between: the near-predictable dynamics which often hold in many industrial applications, e.g., elevator scheduling and bin packing, as two empirical case studies investigated in this paper. Specifically, we propose a two-stage MDP to decouple the state transition uncertainty caused by the data dynamics and constrained action space in the industrial environment. A bi-critic framework is then devised for amortizing the uncertainty and reducing the variance of value estimation according to the two-stage MDP. Experimental results show that our engine can adaptively handle different dynamics data tasks and outperform recent learning-based models and traditional heuristic algorithms.

## 1 Introduction

The advent of Industry 4.0 has put forward demanding requirements for resolving the sequential decision-making task in the industry. The task that involves planning and scheduling has been researched for decades for its commercial value. The planning task, like the bin packing problem (BPP) (Zhao et al., 2020; Zhu et al., 2021; Duan et al., 2022; Zhao et al., 2022; Zhao & Xu, 2022), involves a series of discrete objects under certain constraints to optimize an objective function. While the scheduling task focuses on allocating limited resources to multiple objects to optimize performance indicators under certain constraints, such as the elevator group scheduling problem (EGSP) (Crites & Barto, 1998; Zheng et al., 2013; Wei et al., 2020), the vehicle routing problem (VRP) (Nazari et al., 2018), and the job scheduling problem (JSP) (Chen & Tian, 2019).

With the increase of the problem scale and variety, methods for finding the optimal solution with effectiveness and joint applicability are becoming more and more attractive yet challenging. Traditional solutions in the industry are often rule-based, tuning a score function with expert experience for specific tasks but can hardly be generalized to others. Others try to formulate the tasks as combinatorial optimization problems and then apply heuristics algorithms (often due to its NP-hardness) — such as search algorithm (ELA, 2019; TUR, 2020) and the greedy algorithm (Ramalingam et al., 2017)—but lack real-time response and scalability. Many learning-based methods (Wei et al., 2020; Zhao et al., 2020) are developed, e.g., RL showing its remarkable advantages in sequential decision-making problems. However, emerging learning-based works still often fall behind the industry standards, which can be partly attributed to the lack of real-world training data, and the unavailability of strong simulators to provide rich and realistic data for training.

We consider two aspects for addressing industrial sequential decision-making tasks with RL. The first and mainly addressed issue in our work is to better utilize the character of the environment

dynamics in the industrial pipeline. Existing efforts (Hadoux et al., 2014; Chandak et al., 2020; Chen et al., 2021) assume general non-stationary environments and develop classical RL algorithms to learn structural features of the environment dynamics, including Meta-RL (Xu et al., 2020; Chen et al., 2021) and context detection RL (Padakandla et al., 2020). However, we argue that, in fact, the environment often bears some inherent regularities, and sometimes it is nearly predictable, e.g., in BPP cases, items of similar shape and size usually appear in a batch. The above existing works neglect such potential near-predictable dynamics and leave space for more tailored algorithmic development. Another practical aspect is to strictly obey hard constraints, which are common in the industry for specific reasons, e.g., safety. Although Chen et al. (2021) and Wei & Luo (2021) further consider the Constrained Markov Decision Process (CMDP) (Altman, 1999) and robust constrained Markov decision process (RCMDP) (Russel et al., 2020) for safe RL (Hewing et al., 2020), they tolerate constraints violations and are not up to industry standard. In particular, enforcing hard constraint often increases the high state transition uncertainty (Mao et al., 2018) and further leads to the high variance problem of value estimation. Thus, we argue that the near-predictability must be more carefully considered to mitigate the challenge of the hard constraints.

In this paper, we propose a **D**ynamic-**A**ware and **C**onstraints-**C**onfined (**DACC**) RL framework for industrial sequential decision-making tasks. Unlike previous RL-based efforts for industrial cases that formulate the problem as non-stationary CMDP, we first identify the non-stationary but near-predictable environmental dynamics and reformulate these tasks as a two-stage MDP (Kim et al., 2019) for its potential to distinguish the effects of environment dynamics (exogenous variables) and constrained action space (endogenous variables) in the state transition. Furthermore, the value estimation based on a two-stage MDP reduces the variance of value estimates without introducing bias, as proved by (Mao et al., 2019). Specifically, we design **DACC**, a bi-critic framework for perceiving the dynamics and making decisions under hard industrial constraints with the guidance of heuristic rules, respectively. By estimating the state value in two stages with our bi-critic framework, we reduce the state transition uncertainty and state value estimation variance caused by the mutually adverse effects of dynamic variability and hard constraints. To evaluate our method's effectiveness and generalization, we conduct experiments on two typical industrial sequential decision tasks: 3D bin packing and elevator group scheduling. For the latter case for which there lacks a realistic simulator, we improve the open-source simulator by adding more constraints, business rules, and logic, and (will) release a more realistic one based on our first-hand engagement with top-tier lift manufacturer to benefit the community. **The highlights of this paper are:**

1) Though many sequential decision-making tasks in the industry often require strict constraints, increasing the high state transition uncertainty to challenge RL-based methods. Fortunately, in this paper, we identify that in many cases, the environment is often near-predictable such that it allows for more tailored MDP model development, which is largely ignored by existing methods.

2) We innovatively separate the state transition process of these industrial tasks into two stages. We derive theoretical solutions embodied by a two-stage MDP to the high variance problem of value estimation appearing in the single-stage settings. We then propose a bi-critic framework called Dynamic-Aware and Constraint-Confined (DACC), to capture the regularity of dynamics and makes decisions under hard industrial constraints.

3) We apply our framework to two representative yet challenging real-world cases: 3D bin packing and elevator group scheduling problems. Results show that our methods outperform conventional rule-based and state-of-the-art learning-based models. Further comparisons with the Meta-RL methods verify our framework's superiority in capturing the inherent regularities in these dynamic industrial scenes. Generalization experiments on the untrained data show that the model generalizes well.

## 2 RELATED WORK

Many works extend the Markov decision process model. Constrained Markov decision process (CMDP) (Altman, 1999) is suitable for constrained physical systems, such as avoiding obstacles or unsafe parts in space. Robust Markov decision process (RMDP) Petrik & Russel (2019) is suitable for scenarios where transition probabilities or rewards are unclear. And robust constrained Markov decision process (RCMDP) (Russel et al., 2020) merges both CMDP and RMDP. Time-Dependent MDP (TMDP) (Boyan & Littman, 2000) considers both stochastic state transitions and stochastic, time-dependent action duration. A two-stage MDP task is designed by (Kim et al., 2019) to differentiate the effects of state transition uncertainty and state-space complexity on the brain's arbitration

between model-based and model-free learning. In this paper, we process state transition caused by exogenous (environmental dynamics) and endogenous variables in two stages. RL algorithms are recently devised for environmental dynamics, such as Meta-RL (learn to learn) (Finn et al., 2017; Nagabandi et al., 2018; Xu et al., 2020) and context detection. Meta-RL learns numerous tasks and acquires prior knowledge to learn new tasks faster. The popular Meta-RL models are model-based methods suitable for mainly state-dependent simulation scenarios like maze games rather than input-dependent scenarios to some extent. The context detection RL methods (da Silva et al., 2006; Padakandla et al., 2020; de Oliveira et al., 2006) assess whether the MDP functions have changed based on sequential observations, creating, updating, and selecting one among several partial models of the environment. Whether to generate a new context depends on expert experience, and the model consumes much memory. CASRL (Chen et al., 2021) considers both meta and context-detection. Compared with these works for general dynamics, our efforts are devoted to a tailored approach to better capture the industry dynamics, which are often near predictable.

## 3 PROPOSED APPROACH

We propose the **D**ynamic-**A**ware and **C**onstraint-**C**onfined (**DACC**) RL framework for industrial sequential decision-making tasks under the typical setting: near-predictable dynamics and hard constraints. We first reformulate it as a two-stage MDP and show its inherent properties of reducing state transition uncertainty in § 3.1. In § 3.2, we design our bi-critic framework based on our reformulation, which captures the near-predictable dynamics and confines the hard constraints.

### 3.1 REFORMULATION AS A TWO-STAGE MDP

**Preliminaries.** Many industrial decision-making tasks can be described as a two-stage process: first, the environment generates a new input for decision-making (e.g., a new-coming item to be placed in BPP, a new-coming passenger to be assigned an elevator in EGSP); second, the RL agent takes planning/scheduling actions under real-world constraints for the new input.

**Some Remarks.** As the new input's generation is stochastic while the effect of an action on the state is deterministic after the input's generation, the significant uncertainty of environment dynamics falls into the first stage (i.e., the stochastic generation of the input). The second stage focuses on the uncertainty caused by the restricted action space according to real-world hard constraints. These two sources of uncertainty during the state transition process pose significant challenges of high variance of value estimation (explained in Appendix C) for existing RL algorithms, which formulate these tasks as single-stage MDPs and mix the two sources of uncertainty together. To avoid this problem, we reformulate them as two-stage MDPs and process the two sources of uncertainty separately.

**Notations.** We define a two-stage MDP for industrial sequential decision-making tasks as a tuple $< \mathcal{S}, \mathcal{D}, f, \mathcal{A}, \mathcal{C}, r, \gamma >$, where $\mathcal{S}$ is a finite set of states, $\mathcal{D}$ is a set of the new coming inputs (e.g., new coming items in BPP, new coming passengers in EGSP), $f_t(d)$ is the distribution of the new coming input at time $t$, $\mathcal{A}(a_t|s_t, d_t, \mathcal{C}_t)$ is a finite set of actions under constraints $\mathcal{C}_t$, $r(s_t, a_t, d_t)$ is the reward function, and $\gamma$ is the reward discount factor.

Corresponding to the two-stage process of industrial decision-making tasks, we named the two stages MDP as the input-dependent stage and the state-dependent stage, respectively. Without loss of generality, we use the bin packing problem to describe our approach, especially the two-stage state transition process in Fig. 1. In the input-dependent stage, the generation of new coming input $d_t$ follows a certain time-varying item generating distribution $f_t(d)$, which usually presents task-related and near-predictable patterns over time (i.e., independent of the current state $s_t$). For instance, in many industrial BPP cases, items of similar shape and size usually appear in a batch, thus the distance between $f_t(d)$ and $f_{t+1}(d)$ is small, and the environment dynamics are near-predictable. Hence, the state transition probability of this stage is

$$\mathcal{P}\left(s_t, d_t \mid s_t, f_t(d)\right) = \mathcal{P}\left(d_t \mid f_t(d)\right) = f_t(d) \tag{1}$$

In the state-dependent stage, the constraint $\mathcal{C}_t$ is calculated based on the current state $s_t$ (representing the placement state of previous input items) and the pending current $d_t$. For instance, positions where $d_t$ will topple over are forbidden. Then, the current $d_t$ is picked to the position indicated by the action $a_t$, which the RL agent gives out according to $\left(s_t, d_t, \mathcal{C}_t, f_{t+1:\infty}\right)$, where $f_{t+1:\infty}$ is the features of future item generating distributions that are related to $f_{t+1}(d), f_{t+2}(d), \cdots, f_{t+\infty}(d)$.

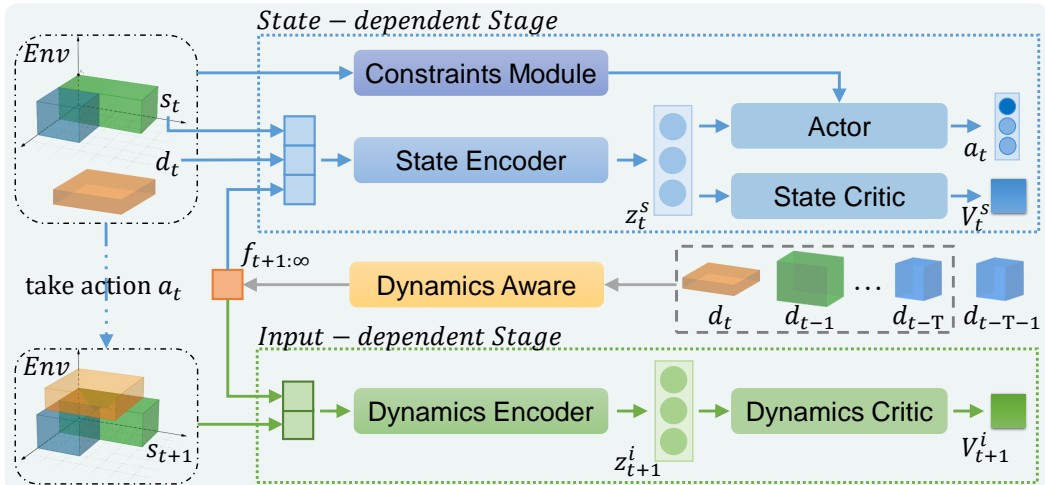

Figure 1: The two-stage state transition process of the BPP. In the input-dependent stage, a new coming item $d_t$ is generated with the current item generating distribution $f_t(d)$. Here $f_t(d)$ denotes the generation probability of three types of items. In the state-dependent stage, the pending $d_t$ is picked to the position indicated by the action $a_t$ under hard constraints $\mathcal{C}_t$.

Figure 2: The proposed bi-critic framework. In the input-dependent stage, the dynamics critic focuses on handling the value estimation based on task-related and time-varying data. In the state-dependent stage, the state critic focuses on handling the value estimation depending on security constraints. And the actor makes decisions based on the learned features.

The transition process of this stage is deterministic such that:

$$\mathcal{P}\left(s_{t+1}, f_{t+1}(d) \mid s_t, d_t, a_t\right) = \begin{cases} 1, & \text{if } a_t \text{ leads to } s_{t+1} \\ 0, & \text{otherwise} \end{cases} \qquad (2)$$

### 3.2 BI-CRITIC FRAMEWORK FOR RL UNDER NEAR-PREDICTABLE ENVIRONMENTS

Based on the reformulation, we design a dynamic-aware and constraints-confined (DACC) reinforcement learning framework for industrial sequential decision-making tasks under near-predictable dynamics. Dealing with the two challenges in two stages separately, we overcome the high variance value estimation problem caused by two sources of state transition uncertainty.

**Value Functions.** In actor-critic-based algorithms (Sutton et al., 1999), for a policy $\pi \in \Pi$, the value function is defined as:

$$V_t(s) := \mathbb{E}[\sum_{\tau \geq t} r(s_\tau, a_\tau, d_\tau) | s_t = s]$$

where the expectation is taken over both the randomness of in policy $\pi$ and the input $d$ ($d_t \sim f_t(d)$).

**Overall Framework.** The overall framework is shown in Fig. 2. For clarity, we plot the state-dependent stage of the current state transition and the input-dependent stage of the next state transition. We use two critic networks $V^s$ and $V^i$ to estimate the state value for the state-dependent stage and input-dependent stage, respectively. We first use the dynamics aware module to extract the latent features $f_{t+1:\infty}$ of future data with historical data $d_t, d_{t-1}, \cdots, d_{t-T}$, where $T$ is the length of historical data used to predict $f_{t+1:\infty}$. Then, for the state-dependent stage, we use the state encoder to encode $\left(s_t, d_t, f_{t+1:\infty}\right)$ and output a latent vector (denoted as $\mathbf{z}_t^s$), which is a multidimensional

Gaussian distribution. After that, we estimate the current state value $V_t^s$ with the state critic network and use the actor network to calculate the action vector $\pi(a_t|\mathbf{z}_t^s)$ under the hard constraints of industrial tasks. For the input-dependent stage, we use the dynamics encoder to extract input-dependent features $\mathbf{z}_{t+1}^i$ (similar to $\mathbf{z}_t^s$) from $s_{t+1}$ and $f_{t+1:\infty}$. Then we use the dynamics critic network to estimate the input-dependent state value $V_{t+1}^i$. We can readily resort to off-the-shelf actor-critic-based algorithms (e.g., A2C as used in our experiments for its simplicity and effectiveness, or others like A3C (Mnih et al., 2016), PPO (Schulman et al., 2017)) to train our model. With the derivation of variance reduction shown in Appendix D, we calculate the advantage function of A2C as

$$A_t = \alpha r_t + \gamma V^i(s_{t+1}, f_{t+1:\infty}) - V^s(s_t, d_t, f_{t+1:\infty}) \tag{3}$$

From the state transition probability of the input-dependent stage (shown in Eq. 1), we get that after both critic networks converge, the relationship between $V^i$ and $V^s$ is as follows:

$$V^i(s_t, f_{t:\infty}) = \sum_{d_t}^{\mathcal{D}} V^s(s_t, d_t, f_{t+1:\infty}) \tag{4}$$

To achieve this and reduce the variance caused by the uncertainty of inputs, we introduce an additional loss function $\mathcal{L}_{KL}$ to minimize the Kullback-Leibler Divergence between $\mathbf{z}_{t+1}^i$ and $\mathbf{z}_t^s$:

$$\mathcal{L}_{KL} = \sum_{i=1}^{k} \left( \log \frac{\sigma_{t+1,(j)}^i}{\sigma_{t,(j)}^s} + \frac{(\sigma_{t,(j)}^s)^2 + \left( \mu_{t,(j)}^s - \mu_{t+1,(j)}^i \right)^2}{2(\sigma_{t+1,(j)}^i)^2} - \frac{1}{2} \right) \tag{5}$$

where $\mu_{t,(j)}^s$ and $(\sigma_{t,(j)}^s)^2$ represent the $j^{th}$ component of the mean vector and variance vector of $\mathbf{z}_t^s$, respectively, while $\mu_{t+1,(j)}^i$ and $(\sigma_{t+1,(j)}^i)^2$ represent those of $\mathbf{z}_{t+1}^i$, respectively, and $k$ is the dimension of $\mathbf{z}_t^s$ and $\mathbf{z}_{t+1}^i$. Thus the overall loss function is defined as

$$\mathcal{L} = A_t{}^2 - A_t \log \pi(a_t|\mathbf{z}_t^s) + \pi(a_t|\mathbf{z}_t^s) \log \pi(a_t|\mathbf{z}_t^s) + \mathcal{L}_{KL} \tag{6}$$

The overall training algorithm is shown in Alg. 1 in Appendix E.

**Dynamics Aware Module.** To capture the trend of data and learn the features of the marginal distribution of future data, we propose a dynamics aware module, which takes into the historical data $d_{t-T}, d_{t-T+1}, \cdots, d_t$ and outputs the feature vector $f_{t+1:\infty}$. As the distribution of data $f_t(d)$ is independent of the current state, we can pre-train the dynamics aware module beforehand. A feasible alternative is to build the dynamics aware module with long short-term memory (LSTM) (Hochreiter & Schmidhuber, 1997) or Attention (Vaswani et al., 2017) networks and set a pre-training task using the last $T$ data to predict the next three data. For instance, we input the data sequence of $d_{t-T}, d_{t-T+1}, \cdots, d_{t-1}$ to the dynamics aware module and make it predict the $d_t, d_{t+1}, d_{t+2}$. After that, we use the pre-trained dynamics aware module to output the hidden feature vector $f_{t+1:\infty}$. The detailed pre-training algorithm is shown in Alg. 2 in Appendix E.

**Constraints and Rules.** To better satisfy strict constraints and make use of task-related heuristic rules in industrial applications, we design a constraints module and heuristic-based reward shaping. For constraints, the technique is the hard action mask. Concretely, it uses task-related constraints to evaluate the feasibility and quality of each action in the current state and the pending input. With its output action mask $\mathbf{M}$, we then get an adjusted policy function: $\pi' \leftarrow \pi(a_t|\mathbf{z}_t) \circ \mathbf{M}$. By explicitly changing the action probability with task-related constraints, we avoid excessive exploration of inefficient or illegal actions (e.g., actions that violate constraints) without introducing excessive training complexity. In practical usage, we can set the chosen probability of illegal actions close to zero to enforce the security constraints. Detailed settings of this module for the two industrial cases can be found in § 4. The other technique we use is heuristic-based reward shaping for its advantage of amortizing the complexity of long-term credit assignment and leveraging heuristic algorithms to accelerate RL's training(Cheng et al., 2021). Heuristic functions that estimate the expected state value are ubiquitous in industrial scenarios. For example, we can use the ETA algorithm to estimate the average waiting time of existing passengers as the heuristic state value $h(s)$ in EGSP. With such heuristic functions, we can introduce heuristic guidance to the RL agent by changing the rewards and lowering the discount factor of the original MDP:

$$\tilde{r}_t(s, d, a) := r_t(s, d, a) + (1 - \lambda)\gamma \mathbb{E}_{s'|s,d,a}[h(s')] \quad \text{and} \quad \tilde{\gamma} := \lambda\gamma, \tag{7}$$

where $\lambda \in [0, 1]$ is the mixing coefficient that increases iteratively along the training process.

## 4 EXPERIMENTS WITH TWO CASE STUDIES

We show the implementations and experiments on two real-world NP-hard industrial scenarios: the bin packing problem (BPP) in § 4.1, and the elevator group scheduling problem (EGSP) in § 4.2.

### 4.1 3D BIN PACKING PROBLEM

**Problem Settings.** The 3D bin packing problem is the essence of mixed palletizing in logistics and warehousing. For a given set of items and containers of a fixed size, the goal is to find the assembly method with the highest space utilization and the least number of containers required. Here we consider the online scenario. Given a sequence of rectangular items $I$ with size $x_i \times y_i \times z_i$ (for each $i \in I$) and a standardized pallet with size $L \times W$ and a maximum height limit $H$, the palletization agent, e.g., a robot, only sees the upcoming item and selects a position to place the item. **Environment Dynamics.** Elhedhli et al. (2019) analyzes the actual industrial data and finds that the characteristics of depth/width ratio, height/width ratio, and repetition follow specific distributions. The average item volume of different industries and categories is different.

**Realistic Constraints.** The placement process is subject to the following constraints as widely used in literature (López-Camacho et al., 2013; Zhao et al., 2022): i) no space overlap between items on the pallet; ii) while sliding a new item to the assigned position, no collision with the items already placed; iii) all placed items are physical of stability.

#### 4.1.1 IMPLEMENTATION COMPONENTS

**MDP.** Specifically, the *state* is represented by the pallet's height, $s_t = \mathbf{H}_t$. The *action* is the position selection for item placing. An invalid action mask $\mathbf{M}_t$ filters out feasible positions and guides exploration. In line with the standard practice in literature, the *reward* $r_t$ for each successful placing step is set proportional to $\frac{x_t \times y_t \times z_t}{L \times W \times H}$, and $r_t = 0$ if the placement fails.

**Network Architecture.** We use attention CNN (CBAM) (Woo et al., 2018) as state encoder, long short-term memory (LSTM) (Hochreiter & Schmidhuber, 1997) as dynamics aware module, and fully-connected networks (FCN) as Critic net and Actor net. Considering that the representation of the pallet is fine-grained and the state dimension is relatively large, we leverage the attention mechanism to increase focus on key regions. The input of the pallet state is a height map $\mathbf{H}_t \in \mathbb{R}^{L \times W}$. And an action mask $\mathbf{M}_t$ represents security constraints for each grid.

Figure 3: Constraints satisfaction for the BPP task. (a) and (b) shows the convex hull calculation, (c) shows a bad case resulting from considering only the convex hull constraint, and (d) shows the four sliding directions we defined.

The size of $\mathbf{H}_t$ and $\mathbf{M}_t$ is doubled while allowing the item to rotate (two rotations: $xyz$ and $yxz$). Since the action mask calculation takes extended time, we decouple the action space into multiple sub-regions to reduce mask calculation. Details of the hierarchical architecture are given in Appendix F.

**Constraints and Rules.** We use an invalid action mask to guarantee two constraints (stability and collision avoidance), a corner point rule to speed up the computation of the invalid action mask and guide the placement of items, and a suitable heuristic to guide placement and speed up algorithm convergence. For stability, we use the convex hull and maintain a solid mask. As Fig. 3(a) and 3(b) show, when placing the yellow item, the centroid of the yellow item must fall within the convex hull of the bottom surface in contact with the lower item. To avoid bad cases like Fig. 3(c) (satisfy that the centroid falls within the convex hull of the contact base but is unstable), we also maintain a solid global mask. Only the intersection of the convex hull and the supporting surface is set to solid, and the item centroid should fall within the solid mask. To avoid collisions, the placement process is to push the item laterally to a particular position so that even if a collision occurs, it will only cause a slight movement of adjacent items. The four sliding directions for selection are shown in Fig. 3(d), including left-back to right-front, left-front to right-back, right-front to left-back, and right-back to left-front. To speed up the invalid action mask calculation, in terms of rules, when selecting actions,

we emphasize the selection of corner points, increase the weight of corner points and only consider sub-regions with corner points, which can significantly reduce the number of solutions, speed up action mask calculation, and improve space utilization (Crainic et al., 2008; Martello et al., 2000; CRA, 2009). To deal with heuristic-based reward shaping, we select the best-performing heuristic on the dataset from four heuristics in the compared baselines to guide the RL agent.

### 4.1.2 EXPERIMENTAL RESULTS

**Datasets and Hyperparameters.** We test our model on two datasets generated differently and containing multiple subsets with different distributions. Pybullet, a physics engine that supports 3D collision detection, has verified the test results. **Mixed-item Dataset (MI Dataset)** is a collection of mixed items with a great variety and few identical items. The data generation scheme follows the realistic 3D-BPP instance generator proposed in (Elhedhli et al., 2019). The training set has 10 million items, with 36,294 species, and occurrences vary from 1 to 7,037. The testing set has 10 thousand items, with 4,886 species, and occurrences vary from 1 to 15. We set pallet dimensions to the size often used in practice: L = 120, W = 100, and H = 100, allowing two packing rotation directions. **Large-item Dataset (LI Dataset)** is a collection of large items and comes from (Martello et al., 2000), including five classes from (Martello & Vigo, 1998) and three class from (Berkey & Wang, 1987). The pallet dimensions are set to L = 100, W = 100, and H = 100, allowing two packing rotation directions. The training set has 39,500 items, with 24,889 species, and occurrences vary from 1 to 12. The testing set has 500 items, with 401 species total, and occurrences vary from 1 to 4. We initialize $\lambda_0 = 0.95$ in Eq. 7 and set $\alpha = 1$, $\gamma = 1$ in Eq. 3 and learning rate $\eta = 1 \times 10^{-5}$.

**Compared Baselines.** We compare our approach with four heuristics and two learning-based methods, presented in Appendix A. For heuristics, we compare with four classic heuristics, which select free spaces represented by Empty Maximum Space (EMS): Bottom-Left heuristic (BL) (Chazelle, 1983), Deepest-Bottom-Left with Fill heuristic (DBLF) (Kang et al., 2012), Best Match First Packing heuristic (BMF) (Li & Zhang, 2015), Online BPH (Ha et al., 2017). As for learning methods, we compare with an outstanding RL BPP model PCT (Zhao & Xu, 2022) in a continuous setting and a model-based Meta-RL model CASRL (Chen et al., 2021). Besides, model A2C (Mnih et al., 2016) (with rules), A2C (w/o rules) and **DACC (w/o rules)** are for ablation study; A2C represents **DACC** without bi-critic architecture and (w/o rules) means no rule guidance.

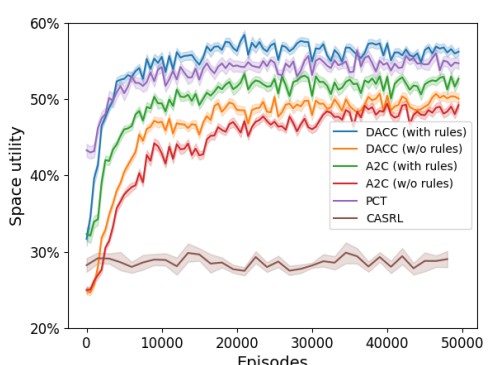

Figure 4: Training rewards of learning-based models on MI Dataset for 3D bin packing.

**Result.** Table 1 shows online bin packing results on two representative datasets. Our model performs well on both mixed-item and large-item datasets, with the flexibility to adapt to different pallet sizes and high-precision placement requirements. Our model achieves higher space utility rates than traditional rule-based algorithms, RL-based models, and the meta-RL model. For learning-based models, the reward curves over training episodes are shown in Fig. 4, whereby the confidence interval is 85. As can be seen from the Fig. 4, our model converges faster than other learning models.

**Ablation Study.** Compared to A2C (without bi-critic structure) without rules, the A2C with rules, and **DACC** without rules, **DACC** can converge to stable performance earlier with higher reward, which means the bi-critic structure and the rule guidance are beneficial. After adding rules (in constraints and rewards), the superiority of **DACC** over A2C is further enlarged, which means that **DACC** can better adapt to rule guidance.

### 4.2 ELEVATOR GROUP SCHEDULING PROBLEM

**Problem Setting.** The elevator group scheduling problem (EGSP) mainly studies assigning elevators to respond to a flood of passengers in buildings equipped with multiple elevators. Among various EGSP systems, we consider a prevalent one: at every time step, stochastic passengers appear and give hall call requests with up/down directions; the system assigns an elevator for every

Table 1: Performance comparison between the baselines and **DACC** in MI Dataset and LI Dataset.

| Method | MI Dataset | | LI Dataset | |
|---|---|---|---|---|
| | Space utility ↑ | Avg. items ↑ | Space utility ↑ | Avg. items ↑ |
| **BL** (Chazelle, 1983) | 22.55% | 9.32 | 27.97% | 2.04 |
| **DBLF** (Kang et al., 2012) | 26.54% | 10.51 | 27.18% | 1.98 |
| **BMF** (Li & Zhang, 2015) | 29.67% | 11.47 | 27.28% | 1.99 |
| **Online BPH** (Ha et al., 2017) | 24.56% | 10.82 | 24.77% | 2.18 |
| **CASRL** (Chen et al., 2021) | 11.18% | 3.75 | 17.92% | 2.22 |
| **PCT-continues** (Zhao & Xu, 2022) | 56.59% | 19.50 | 41.18% | 3.22 |
| **A2C (w/o rules)** (Mnih et al., 2016) | 48.89% | 16.53 | 37.04% | 2.97 |
| **A2C (with rules)** (Mnih et al., 2016) | 55.24% | 19.15 | 40.77% | 3.11 |
| **DACC (w/o rules)** | 54.55% | 18.40 | 38.71% | 3.03 |
| **DACC** | **58.37%** | **19.75** | **43.49%** | **3.25** |

hall call request without knowing the passengers' destination floors until they get on the elevator; also, passengers do not know the assigned elevators so that they can be reassigned at any time.

**Environment Dynamics.** Passenger flow in the EGSP often shows specific traffic daily patterns, which can be divided into three major periods (Jansson & Uggla Lingvall, 2015): up-peak traffic in morning rush hours, two-way traffic at lunchtime, and down-peak traffic in evening rush hours.

**Realistic Constraints.** There are some commonly-used constraints for the elevator assignment and movement. 1) The EGSP system is required to dispatch exactly one elevator for every hall call request; 2) The assigned elevator must respond to the hall call request while completing the car calls generated by passengers on it (a car call indicates the passenger's destination floor and is registered at the elevator after the passenger enters it; 3) The elevator can not change its service direction until there is no hall call or car call in the current direction.

### 4.2.1 IMPLEMENTATION COMPONENTS

**Simulator.** To mimic the real-world behavior of EGSP, based on the only available open-source simulator Liftsim (Wang et al., 2020), we further improve it into a more realistic EGSP simulator that satisfies the three realistic constraints mentioned in the last paragraph. Additionally, our simulator can simulate passenger flow following the three traffic patterns. Detailed comparisons among existing simulators are explained in Appendix J.

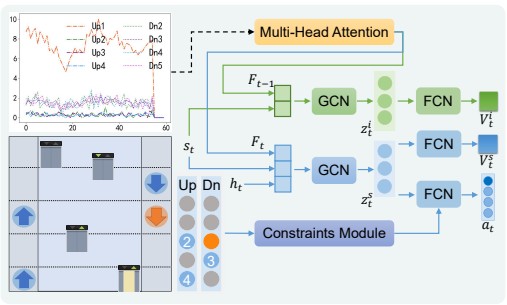

Figure 5: DACC of EGSP. We use a multi-head attention network to extract features from historical hall call data, graph convolutional networks (GCN) as state and dynamics encoders, and fully-connected networks (FCN) as critics and the actor.

**MDP.** The simulator formulates EGSP as a two-stage MDP: in every simulation time step, new passengers are generated with a probability matrix corresponding to the specific pattern; the agent receives the state information from the simulator and makes a decision to assign elevators for existing unprocessed hall calls. As shown in Fig. 4.2, the state includes the elevator state $s_t$ and the hall call state $h_t$. The elevator state includes the position, speed, service direction, door status, and registered car call of all elevators. And the hall call state includes the unprocessed requests (marked orange in Fig. 4.2), the existing hall call requests, and their corresponding assigned elevators. The action is defined as assigning an elevator for each unprocessed hall call, and the objective is to reduce the average waiting time (denoted as $T_{aw}$) and the average transmitting time (denoted as $T_{at}$) of all passengers. At the same time save energy consumption. Here the waiting time of a passenger is the time taken from the moment he presses the hall call button on a floor to the instant he is picked up. The transmitting time is the time taken from when one is picked up to the instant one reaches the destination floor. Thus the reward function is defined as

$$r_t = -(\beta|N_q| + \xi|N_c| + E), \tag{8}$$

where $N_q, N_c$ is the number of waiting passengers in the hall, and those in the elevators, respectively, as weighted by hyperparameter $\beta, \xi$. $E$ is the energy consumption measured by electricity cost.

Table 2: Evaluation of the baselines and **DACC** in three traffic patterns in terms of two criteria: $T_{as}$ (the sum of $T_{aw}$ and $T_{aw}$) and $TE$ (a criteria that integrates $T_{as}$ and energy consumption $E$).

| Method | Two-way | | Up-peak | | Down-peak | |
|---|---|---|---|---|---|---|
| | $T_{as}\downarrow$ | $TE\downarrow$ | $T_{as}\downarrow$ | $TE\downarrow$ | $T_{as}\downarrow$ | $TE\downarrow$ |
| **ETA** (Rong et al., **2003**) | 92.50 | 110.98 | 119.83 | 142.47 | **65.30** | 70.50 |
| **SFM** (Ramalingam et al., **2017**) | 90.99 | 110.67 | 119.27 | 141.98 | 70.63 | 75.37 |
| **Robert** (Crites & Barto, **1998**) | 159.31 | 261.70 | 126.70 | 247.87 | 177.42 | 209.21 |
| **DRL-EGC** (Wei et al., **2020**) | 112.93 | 134.89 | 123.60 | 151.39 | 82.21 | 86.40 |
| **Context-QL** (Padakandla et al., **2020**) | 91.14 | 111.18 | 114.89 | 137.03 | 66.34 | 71.82 |
| **DACC** | **89.75** | **108.14** | **110.21** | **132.15** | 65.83 | **70.35** |

**Network Architecture.** As shown in Fig. 4.2, we use a multi-head attention network (Vaswani et al., 2017) as the dynamics aware module to capture the trend of passenger flow and learn the hidden features from historical hall call data. We record the number of generated hall calls on each floor every minute during the whole simulation process. For the current time step $t$, we forward the collected hall call data in the last 30 minutes to the attention network and use a two-layer FC network to output the latent feature vector distribution $f_{t+1:\infty}$. We use two four-layer graph convolutional networks (GCN) (Kipf & Welling, 2016) as the state encoder and the dynamics encoder, respectively. And we use two-layer FC networks as the critic networks and the actor-network, respectively.

**Constraints and Rules.** We develop an action mask function to avoid choosing actions that violate the realistic constraints mentioned above. And we incorporate the estimated time of arrival algorithm (ETA) as a heuristic rule to guide the exploration of the RL agent, which can reduce the occurrence of inefficient actions such as assigning an elevator running oppositely to a hall call.

### 4.2.2 EXPERIMENTAL RESULTS

**Dataset and Hyperparameters.** We use passenger flow data in 30 days respectively for two-way traffic, up-peak, and down-peak patterns, provided by a world-renowned elevator manufacturer, from a real-world 16-floor and 4-elevator building, with nearly 50 persons residing on each floor except the first floor. For each pattern, we calculate a probability matrix $\mathbf{P}_t$ that represents the generating probability of a passenger who appears at floor $i$ and targets floor $j$ at time $t$ and use $\mathbf{P}_t$ to generate passengers randomly. We initialize $\lambda_0 = 0.95$ in Eq. 7 and set $\alpha = 0.01$, $\gamma = 0.99$ in Eq. 3, $\beta = 1$, $\xi = 0.6$ in the reward Eq. 8, and the learning rate as $\eta = 2 \times 10^{-5}$.

**Compared Baselines.** We compare our model with traditional EGSP algorithms, including Round-Robin, estimated time of arrival (ETA) (Rong et al., 2003), and genetic algorithm (GA), among which ETA empirically achieves the best performances in most cases. The SFM (Ramalingam et al., 2017) algorithm for EGSP is a combinatorial optimization-based algorithm known to be adopted by commercial elevator companies. Learning-based algorithms for EGSP include Robert (Crites & Barto, 1998) (using Q-Learning) and DRL-EGC (Wei et al., 2020) (using the A3C (Mnih et al., 2016) algorithm). Additionally, we train a context detection method for non-stationary environments Context Q-learning (Padakandla et al., 2020) in our environment as a baseline.

**Results.** As shown in Table 2, our framework outperforms all existing methods for EGSP in all of the three traffic patterns in terms of the $TE$ criteria, which takes the average waiting time $T_{at}$, the average transmitting time $T_{at}$ and energy consumption $E$ into consideration. For the $T_{as}$ criteria, which only considers the average waiting time and transmitting time, our framework outperforms all existing methods in the two-way traffic and up-peak patterns and achieves comparable performance with the best algorithm in the dn-peak pattern.

## 5 CONCLUSION

Identifying the near-predictability of many industrial dynamic environments, we have devised a tailored RL-based framework using bi-critic to handle the typical industrial sequential decision-making tasks with two successful case studies for 3D bin packing and elevator group scheduling. We also develop a more realistic simulator compared with current open-source simulators.

**Limitation & Future Work.** As our approach is tailored to the near-predictable setting, it may fail in cases when the dynamics are more uncertain, as shown in our failure case study in Appendix I.

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

# A    ADDITIONAL RELATED WORKS

**Rule-based Algorithms for Sequential Decision-Making Tasks.**  Traditional practices usually utilize rule-based algorithms with expert experience for specific tasks. For BPP, the Bottom-Left heuristic (BL) (Chazelle, 1983) searches for Empty Maximum Space (EMS) with the minimum bottom ($x$) and left ($y$) coordinates to place the current item; Deepest-Bottom-Left with Fill heuristic (DBLF) (Kang et al., 2012) sorts EMS in order of deepest coordinates ($z$) and uses bottom and left as tie-breakers; Best Match First Packing heuristic (BMF) (Li & Zhang, 2015) sorts EMS in order of distance to the container's deepest-bottom-left point; and Online BPH (Ha et al., 2017) sorts EMS in order of minimum bottom, deepest, and minimum left. For EGSP, collective control (Strakosch, 1983) assigns the elevator to stop at its nearest call in the running direction; and (Rong et al., 2003) allocates hall calls to the elevator that minimizes the estimated arrival time. Others create policies with matrices of "heuristic scores" and option decisions with the highest score. (Ramalingam et al., 2017) develops a greedy algorithm based on submodularity to maximize its objective function.

**Learning-based Algorithms for Sequential Decision-Making Tasks.**  Reinforcement learning (RL) excels at dealing with sequential decision-making problems. In real-world BPP applications, the agent usually can not foresee the upcoming items, which puts online decision requirements on BPP solvers. This variant of BPP is called online BPP, and Zhao et al. (2020) use RL and well-designed discrete representations to solve it. While Zhao & Xu (2022) treat the BPP problem as a decision in a continuous space and formulate it into a tree structure. The model combines RL and traditional algorithms for flexible applications in various settings. As for EGSP, an early work (Crites & Barto, 1998) uses Q-learning to control elevators to move up and down. A recent work (Wei et al., 2020) explores deep asynchronous actor-critic learning to decide each elevator's next target stop floor. Our model focuses on input-dependent industrial scenarios, where input data are collected from production line with regularity to follow.

# B    COMPARISON WITH EXISTING RL MODELS WITH EXOGENOUS INPUTS

Recent works investigate RL with exogenous inputs. (Mao et al., 2019) proposes a meta-RL model that formally defines input-driven MDP and proves that an input-dependent baseline is bias-free and variance-reduced when input and action are conditionally independent. Our theoretical basis is the same as (Mao et al., 2019): considering the sequence of input values in value estimation can reduce estimation variance without introducing bias. Different from (Mao et al., 2019), to deal with a class of industrial problems, we consider the input and constraints in two stages during the state transition, thereby designing a bi-critic network. Chitnis & Lozano-Pérez (2020) divides the state $s_t$ into endogenous component $n_t$ and exogenous component $x_t$. Then split exogenous component $x_t$ into $m$ state variables $x_t^1, x_t^2, \ldots, x_t^m$ and learn a mask, a subset of the exogenous state variables. Chitnis & Lozano-Pérez (2020)'s decomposition of state transition is similar to ours, but the subsequent learning is completely different. Sinclair et al. (2022) samples trajectories over the exogenous trace from the history dataset and trains the model-based RL model offline. Given a fixed exogenous sequence $\boldsymbol{\xi} = \{\xi_1, \ldots, \xi_T\}$ and policy $\pi$, Sinclair et al. (2022) decomposes the value function into an expectation of a fixed exogenous trajectory $V_t^\pi(s, \boldsymbol{\xi}_{\geq t}) := \sum_a \pi(a \mid s) Q_t^\pi(s, a, \boldsymbol{\xi}_{\geq t})$. The training method and handling of value function are not the same as ours.

# C    DERIVATION OF REDUCING THE VARIANCE OF VALUE ESTIMATION CAUSED BY MIXING TWO SOURCES OF UNCERTAINTY

Previous works Mao et al. (2019) prove that subtracting an action-independent baseline function help reduce value estimation variance for actor-critic-based algorithms. However, the problem of high variance of value estimation is still exposed in industrial decision-making tasks because the two sources of uncertainty during the state transition process are mixed and processed together (shown in Fig. 6) by existing RL algorithms. Here we use the case of BPP to illustrate why. The same is true for other tasks. Without loss of generality to all actor-critic-based algorithms, here we use the advantaged function of the A2C algorithm for illustration:

$$A_t = r_t + \gamma V_{t+1} - V_t \tag{9}$$

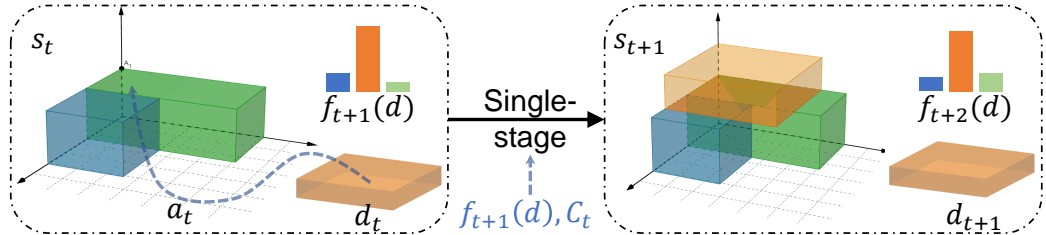

Figure 6: The single-stage state transition process of the BPP. It takes $(s_t, d_t, f_{t+1})$ as the state in regular MDP formulation. And during the state transition process, it mixes the two sources of uncertainty ($f_{t+1}(d)$ and $\mathcal{C}_t$), leading to a high variance of value estimation.

Consider the state transition process from $(s_t, d_t, f_{t+1})$ to $(s_{t+1}, d_{t+1}, f_{t+2})$ in the single-stage formulation, where state $s_t$ at time $t$ denotes the current placing state, $d_t$ denotes the current pending item, and $f_{t+1}$ denotes the generation distribution of the next coming item $d_{t+1}$. Additionally, we use $f_{t+1:\infty}$ to denote the marginal distribution of future items at time $t$, which is determined by $f_{t+1}, f_{t+2}, \cdots, f_{t+\infty}$. And we denote the finite set of item types as $\mathcal{D}$, the finite action set as $\mathcal{A}$, the policy as $\pi_\theta$, hard constraints at time $t$ as $\mathcal{C}_t$, the reward function as $r$ and the discount factor as $\gamma$. Suppose that the value estimation function inputs at time $t$ are the $s_t$, $d_t$, and $f_{t+1:\infty}$. The advantage function can be written as

$$A_t = r_t + \gamma V(s_{t+1}, d_{t+1}, f_{t+2:\infty}) - V(s_t, d_t, f_{t+1:\infty})] \tag{10}$$

During the training process of the A2C algorithm with plenty of exploration, $A_t$ is approximate to its expectation:

$$\begin{aligned}
\mathbb{E}_\pi[A_t] &= \mathbb{E}_\pi[r_t + \gamma V(s_{t+1}, d_{t+1}, f_{t+2:\infty}) - V(s_t, d_t, f_{t+1:\infty})] \\
&= \sum_{d_t}^{\mathcal{D}} f_t(d_t) \sum_{a_t}^{\mathcal{A}} \pi_\theta(a_t|s_t, d_t, \mathcal{C}_t) \sum_{d_{t+1}}^{\mathcal{D}} f_{t+1}(d_{t+1})[r(s_t, d_t, a_t)+ \\
&\quad \gamma V(s_{t+1}, d_{t+1}, f_{t+2:\infty}) - V(s_t, d_t, f_{t+1:\infty})]
\end{aligned} \tag{11}$$

For simplicity, we define the hard constraint as a function prohibiting actions that cause the pending item to tip over:

$$\mathcal{C}_t(s_t, d_t, a_t) = \begin{cases} 0, & \text{if } a_t \text{ makes } d_t \text{ tip over at } s_t \\ 1, & \text{otherwise} \end{cases} \tag{12}$$

Then we can simplify the policy function:

$$\pi_\theta(a_t|s_t, d_t, \mathcal{C}_t) = \pi_\theta(a_t|s_t, d_t)\mathcal{C}_t(s_t, d_t, a_t) \tag{13}$$

And we can further get

$$
\begin{aligned}
\mathbb{E}_\pi[A_t] = & \sum_{d_t}^{\mathcal{D}} f_t(d_t) \sum_{a_t}^{\mathcal{A}} \pi_\theta(a_t|s_t, d_t) \mathcal{C}_t(s_t, d_t, a_t) \sum_{d_{t+1}}^{\mathcal{D}} f_{t+1}(d_{t+1})[r(s_t, d_t, a_t)+ \\
& \gamma V(s_{t+1}, d_{t+1}, f_{t+2:\infty}) - V(s_t, d_t, f_{t+1:\infty})] \\
= & \sum_{d_t}^{\mathcal{D}} f_t(d_t) \sum_{a_t}^{\mathcal{A}} \pi_\theta(a_t|s_t, d_t) \mathcal{C}_t(s_t, d_t, a_t) r(s_t, d_t, a_t) + \sum_{d_t}^{\mathcal{D}} f_t(d_t) \sum_{a_t}^{\mathcal{A}} \pi_\theta(a_t|s_t, d_t) \\
& \mathcal{C}_t(s_t, d_t, a_t) \sum_{d_{t+1}}^{\mathcal{D}} f_{t+1}(d_{t+1})[\gamma V(s_{t+1}, d_{t+1}, f_{t+2:\infty}) - V(s_t, d_t, f_{t+1:\infty})] \\
= & \sum_{d_t}^{\mathcal{D}} f_t(d_t) \sum_{a_t}^{\mathcal{A}} \pi_\theta(a_t|s_t, d_t) \mathcal{C}_t(s_t, d_t, a_t) r(s_t, d_t, a_t)+ \\
& \gamma \sum_{d_t}^{\mathcal{D}} f_t(d_t) \sum_{a_t}^{\mathcal{A}} \pi_\theta(a_t|s_t, d_t) \mathcal{C}_t(s_t, d_t, a_t) \sum_{d_{t+1}}^{\mathcal{D}} f_{t+1}(d_{t+1}) V(s_{t+1}, d_{t+1}, f_{t+2:\infty})- \\
& \sum_{d_t}^{\mathcal{D}} f_t(d_t) \sum_{a_t}^{\mathcal{A}} \pi_\theta(a_t|s_t, d_t) \mathcal{C}_t(s_t, d_t, a_t) V(s_t, d_t, f_{t+1:\infty})]
\end{aligned}
\tag{14}
$$

We focus on the second term of Eq. 14. Notice that the generation of $d_{t+1}$ is stochastic and follows the distribution of $f_{t+1}(d_{t+1})$, which is unknown to the RL agent. Thus during each training episode, the term $\gamma V(s_{t+1}, d_{t+1}, f_{t+2:\infty})$ in Eq. 10 introduces a lot of variance due to the uncertainty of $d_{t+1}$. Moreover, the constraint term $\mathcal{C}_t(s_t, d_t, a_t)$ applied to the policy function introduces variance due to the uncertainty of constraints. As the second term in Eq. 14 multiply $\mathcal{C}_t(s_t, d_t, a_t)$ by $V(s_{t+1}, d_{t+1}, f_{t+2:\infty})$, it brings mutually adverse effect on variance and further confuse the RL agent during the training process, hindering the critic network from estimating the state value accurately and the actor-network from learning good policies.

## D   DERIVATION OF VALUE REDUCTION OF THE BI-CRITIC FRAMEWORK

In the two-stage MDP formulation, we use $V^s$ and $V^i$ for value estimation of the state-dependent stage and the input-dependent stage, respectively. In the state-dependent stage, as the state transition process is deterministic (from Eq. 2) and does not care about the generation of $d_{t+1}$, the advantaged function $A_t$ is calculated by

$$
A_t = r_t + \gamma V^i(s_{t+1}, f_{t+1:\infty}) - V^s(s_t, d_t, f_{t+1:\infty})
\tag{15}
$$

Thus the expectation of $A_t$ with sufficient exploration during the training process is calculated by

$$
\begin{aligned}
\mathbb{E}_\pi[A_t] &= \mathbb{E}_\pi[r_t + \gamma V^i(s_{t+1}, f_{t+1:\infty}) - V^s(s_t, d_t, f_{t+1:\infty})] \\
&= \sum_{d_t}^{\mathcal{D}} f_t(d_t) \sum_{a_t}^{\mathcal{A}} \pi_\theta(a_t|s_t, d_t, \mathcal{C}_t)[r(s_t, d_t, a_t) + \gamma V^i(s_{t+1}, f_{t+1:\infty}) - V^s(s_t, d_t, f_{t+1:\infty})] \\
&= \sum_{d_t}^{\mathcal{D}} f_t(d_t) \sum_{a_t}^{\mathcal{A}} \pi_\theta(a_t|s_t, d_t)\mathcal{C}_t(s_t, d_t, a_t)r(s_t, d_t, a_t) + \sum_{d_t}^{\mathcal{D}} f_t(d_t) \sum_{a_t}^{\mathcal{A}} \pi_\theta(a_t|s_t, d_t) \\
&\quad \mathcal{C}_t(s_t, d_t, a_t)[\gamma V^i(s_{t+1}, f_{t+1:\infty}) - V^s(s_t, d_t, f_{t+1:\infty})] \\
&= \sum_{d_t}^{\mathcal{D}} f_t(d_t) \sum_{a_t}^{\mathcal{A}} \pi_\theta(a_t|s_t, d_t)\mathcal{C}_t(s_t, d_t, a_t)r(s_t, d_t, a_t) + \\
&\quad \gamma \sum_{d_t}^{\mathcal{D}} f_t(d_t) \sum_{a_t}^{\mathcal{A}} \pi_\theta(a_t|s_t, d_t)\mathcal{C}_t(s_t, d_t, a_t)V^i(s_{t+1}, f_{t+1:\infty}) - \\
&\quad \sum_{d_t}^{\mathcal{D}} f_t(d_t) \sum_{a_t}^{\mathcal{A}} \pi_\theta(a_t|s_t, d_t)\mathcal{C}_t(s_t, d_t, a_t)V^s(s_t, d_t, f_{t+1:\infty})]
\end{aligned}
\tag{16}
$$

Comparing Eq. 14 and Eq. 16, we get that the major difference is in the second term, where we substitute $V^i(s_{t+1}, f_{t+1:\infty})$ for $\sum_{d_{t+1}}^{\mathcal{D}} f_{t+1}(d_{t+1})V(s_{t+1}, d_{t+1}, f_{t+2:\infty})$ in our two-stage MDP formulation. In other words, we expect to reduce the variance by updating the advantaged function with the expectation of $V(s_{t+1}, d_{t+1}, f_{t+2:\infty})$ (i.e., $V^i(s_{t+1}, f_{t+1:\infty})$ in Eq. 15) instead of the exact $V(s_{t+1}, d_{t+1}, f_{t+2:\infty})$ (in Eq. 10) during the training process. And this goal is achieved by two independent stages in our bi-critic framework. First, the term $\mathcal{C}_t(s_t, d_t, a_t)V^i(s_{t+1}, f_{t+1:\infty})$ does not bring excess variance caused by mixing the two sources of uncertainty. Second, we introduce an additional loss function to minimize the distance between $V^i(s_{t+1}, f_{t+1:\infty})$ and $\sum_{d_{t+1}}^{\mathcal{D}} f_{t+1}(d_{t+1})V^s(s_{t+1}, d_{t+1}, f_{t+2:\infty})$ in the training process, so that we reduce the value estimation variance caused by the uncertainty of $d_{t+1}$.

Hence, our bi-critic framework under the two-stage MDP formulation can reduce value estimation variance, promoting the convergence of the critic network and guiding the actor-network to learn better policies stably.

## E  TRAINING ALGORITHMS

The overall training algorithm for the whole framework is shown in Alg. 1. Note that the input data distribution sequence $f_0(d_0), f_1(d_1), \cdots, f_N(d_N)$ is only for the generation of input data $d_t$ and is unknown to the RL agent, which follows the setting of most industrial sequential decision-making scenarios.

---
**Algorithm 1** Training algorithm for our bi-critic framework
---
Input a task-related heuristic function $\mathcal{H}$;
Input a task-related data distribution sequence $f_0(d), f_1(d), \cdots, f_N(d)$;
Initialize bi-critic networks $V_{\theta_i}^i, V_{\theta_s}^s$, and actor $\pi_\omega$ with random parameters $\theta_i, \theta_s, \omega$;
Initialize state encoder $E_{\phi_i}^s$, dynamics encoder $E_{\phi_s}^i$, dynamics aware module $\mathcal{F}_\psi$, and the constraint module $\mathcal{M}_\zeta$ with random parameters $\phi_i, \phi_s, \psi, \zeta$;
Initialize input buffer $\mathcal{D}$;
Pretrain the dynamics module $\mathcal{F}_\psi$ with Alg. 2;
**for** $episode = 0$ **to** $max\_episode$ **do**
    Reset the environment and observe state $s_0$, reset $\mathcal{D}$;
    Generate $d_0$ with $f_0(d_0)$ and store $d_0$ in $\mathcal{D}$;
    Update $\lambda$: $\lambda \leftarrow \lambda_0 + (1 - \lambda_0) * episode/max\_episode$;
    **for** $t = 0$ **to** $N - 1$ **do**
        # state dependent stage
        Get future data features $f_{t+1:\infty} = \mathcal{F}_\psi(\mathcal{D})$;
        Get the constrained action mask $\mathbf{M}_t = \mathcal{M}_\zeta(s_t, d_t)$;
        Get $\mathbf{z}_t^s = E_{\phi_s}^s(s_t, d_t, f_{t+1:\infty})$, $V_t^s = V_{\theta_s}^s(\mathbf{z}_t^s)$, action $a_t \sim \pi_\omega(a_t|\mathbf{z}_t^s)) \circ \mathbf{M}_t$;
        Perform action $a_t$ in the environment and then observe reward $r_t$ and next state $s_{t+1}$;

        # input dependent stage
        Get $\mathbf{z}_{t+1}^i = E_{\phi_i}^i(s_{t+1}, f_{t+1:\infty})$, $V_{t+1}^i = V_{\theta_i}^i(\mathbf{z}_{t+1}^i)$;
        Generate $d_{t+1}$ with $f_{t+1}(d)$ and store $d_{t+1}$ in $\mathcal{D}$;

        # compute loss and update parameters
        Get advantage estimate with Eq. 3 and Eq. 7:
        $A_t = \alpha r_t + (1 - \lambda)\gamma\mathcal{H}(s_t, d_t, a_t)] + \lambda\gamma V_{t+1}^i - V_t^s$;
        Update all parameters with loss function in Eq. 6;
---

Here we illustrate a simple example for pre-training the dynamics module, which makes it predict the next three input data with previous $T$ input data at time $t$. To ensure the diversity of the input data distributions, we add noise $\mathcal{N}(0, 0.1)$ to them in each training iteration. The algorithm is shown in Alg. 2.

---
**Algorithm 2** Pretrain algorithm for the dynamics aware module
---
Input a task-related data distribution sequence $f_0(d), f_1(d), \cdots, f_N(d)$;
Initialize the dynamics module $\mathcal{F}_\psi$ and a predict network $p_\varphi$ with random parameters $\psi, \varphi$;
Initialize memory buffer $\mathcal{B}$;
**for** $iter = 1$ **to** $max\_iter$ **do**
    Perturb the input data distribution for each time step $t$: $f_t'(d) \leftarrow f_t(d) + \mathcal{N}(0, 0.1)$;
    **for** $t = T - 1$ **to** $N - 3$ **do**
        Reset $\mathcal{B}$;
        **for** $i = 0$ **to** $K$ **do**
            Generate $X_i = d_{t-T+1}, \cdots, d_t$ with $f_{t-T+1}(d), \cdots, f_t(d)$;
            Generate $Y_i = d_{t+1}, d_{t+2}, d_{t+3}$ with $f_{t+1}(d), f_{t+2}(d), f_{t+3}(d)$;
            Store $(X_i, Y_i)$ in $\mathcal{B}$;
            Sample a batch $< X, Y >$ from $\mathcal{B}$;
            Get $f_{t+1:\infty} = \mathcal{F}_\psi(X), \hat{Y} = p_\varphi(f_{t+1:\infty})$;
            Compute the loss function $\mathcal{L} = Cross\_Entropy(Y, \hat{Y})$;
            Update $\psi, \varphi$ with $\mathcal{L}$;
---

## F    BPP HIERARCHICAL ARCHITECTURE

The framework of the BPP placement planner is a two-level hierarchical Actor-Critic with an attention mechanism. Facing the Exploding action space, we decompose the action space (whole pallet)

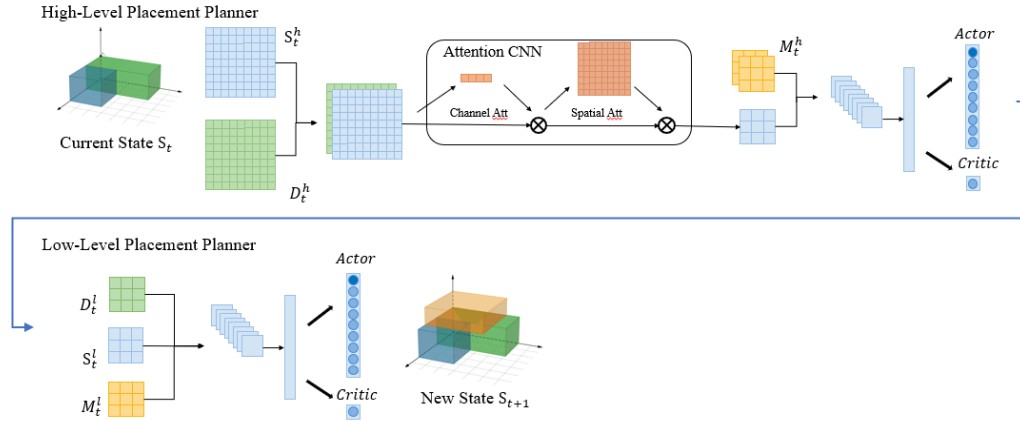

Figure 7: Overview of the proposed hierarchical architecture for the BPP task. Since invalid action mask calculation takes extended time and is essential for decision constraints, we need to decouple the action space to reduce mask calculation. With hierarchical architecture, the entire space of the pallet (9*9) is divided into small regions (3*3), the high-level agent selects a feasible subspace for the low-level, and the low-level agent selects a specific position to pack. Decoupling of action space reduces the size of invalid action masks (colored in yellow), thus reducing computation time.

Table 3: Distribution Functions and Parameters for Item Characteristics

| Characteristic | Distribution | Parameters |
|---|---|---|
| Depth/width ratio | Normal (loc, scale) | $(0.695, 0.118)$ |
| Height/width ratio | Lognormal (mean, sigma) | $(-0.654, 0.453)$ |
| Repetition | Lognormal (mean, sigma) | $(0.544, 0.658)$ |
| Volumes | Lognormal (mean, sigma) | $(2.568 \times V_{cat}, 0.705)$ |
| $V_{cat}$ | Normal (loc, scale) | $(2.5, 0.118)$ |
| $Number_{cat}$ | Normal (loc, scale) | $(1000, 0.118)$ |
| Width clip | [min width, max width] | $[20, 60]$ |
| Depth clip | min depth, max depth | $[20, 50]$ |
| Height clip | min height, max height | $[10, 50]$ |
| $V_{cat}$ clip | min $V_{cat}$, max $V_{cat}$ | $[1, 4]$ |

Table 4: The uniform distribution interval of the items' width, height and depth of each type/class.

| Type/Class | Width | Height | Depth |
|---|---|---|---|
| Type 1 | $\left[1, \frac{1}{2}W\right]$ | $\left[\frac{2}{3}H, H\right]$ | $\left[\frac{2}{3}D, D\right]$ |
| Type 2 | $\left[\frac{2}{3}W, W\right]$ | $\left[1, \frac{1}{2}H\right]$ | $\left[\frac{2}{3}D, D\right]$ |
| Type 3 | $\left[\frac{2}{3}W, W\right]$ | $\left[\frac{2}{3}H, H\right]$ | $\left[1, \frac{1}{2}D\right]$ |
| Type 4 | $\left[\frac{1}{2}W, W\right]$ | $\left[\frac{1}{2}H, H\right]$ | $\left[\frac{1}{2}D, D\right]$ |
| Type 5 | $\left[1, \frac{1}{2}W\right]$ | $\left[1, \frac{1}{2}H\right]$ | $\left[1, \frac{1}{2}D\right]$ |
| Class 6 | $[10, W]$ | $[10, H]$ | $[10, D]$ |
| Class 7 | $\left[1, \frac{7}{8}W\right]$ | $\left[1, \frac{7}{8}H\right]$ | $\left[1, \frac{7}{8}D\right]$ |
| Class 8 | $[1, W]$ | $[1, H]$ | $[1, D]$ |

into a certain number of sub-regions. The high-level policy selects a sub-region with an attention mechanism for the low level to achieve, and the low level selects a specific packing position in the sub-region. As the action space decreases, the corresponding mask also decreases, and the amount of calculation becomes smaller.

# G    DATA GENERATION OF BPP

**Mixed-item Dataset (MI Dataset)** follows the realistic 3D-BPP instance generator proposed in (Elhedhli et al., 2019). The pallet dimensions are set to L = 120, W = 100, and H = 100. The item characteristics are determined by fitting the real-life industry data from (Elhedhli et al., 2019). We gen-

Table 5: Result on the untrained dataset with different location (loc) parameters of $Number_{cat}$'s distribution and scale parameter of $V_{cat}$'s distribution.

| loc of $Number_{cat}$ | scale of $V_{cat}$ | Space utility ↑ | Avg.items ↑ |
|---|---|---|---|
| 500 | 0.118 | 52.36% | 18.44 |
| 100 | 0.118 | 52.79% | 19.00 |
| 10 | 0.118 | 53.91% | 19.32 |
| 100 | 0.5 | 52.72% | 19.93 |
| 100 | 1.0 | 52.78% | 21.57 |

erate data considering the ratio of item depth and height to its width (depth/width and height/width ratio), volume, and frequency of occurrence (Repetition). The first two characterize the size of the item. The reason to consider the proportions and volumes of an item rather than its dimensions is to construct items of different sizes whose dimensions are appropriately related to each other. In real-life industrial environments, the average volume of incoming items varies with each category. In order to simulate the volume changes of different categories, we generate the number ($Number_{cat}$) and average volume ($V_{cat}$) of each category of items according to the normal distribution and then generate the item volume (Volumes) within the category according to the log-normal distribution.

**Large-item Dataset (LI Dataset)** comes from (Martello et al., 2000) with eight classes of data, which generates three-dimensional instances from randomly generated width, depth, and height within a preset interval. The pallet dimensions are set to L = 100, W = 100, and H = 100. The first five classes of data come from (Martello & Vigo, 1998). We generate five types of instances as Table 4. For Class $k(k = 1, \ldots, 5)$, each item is of type $k$ with probability 60%, while it is of the other four types with probability 10% each. The last three classes are a generalization of the instances presented by (Berkey & Wang, 1987).

## H GENERALIZATION EXPERIMENT ON BPP

Since our MI Dataset is built on certain distributions for item characteristics, we can also evaluate the performance of **DACC** on different distributions from training data. First, we change the location parameter of $Number_{cat}$'s distribution and the scale parameter of $V_{cat}$'s distribution. The testing data includes 10000 items generated from different distributions. The result is presented in Table 5. Our method can still perform well while testing on the untrained dataset.

## I FAILURE CASE STUDY

We increase the uncertainty and dynamic of data by changing some data generation parameters of the MI dataset. Then, we train **DACC** on datasets with different uncertainty. To achieve greater uncertainty and minimize the impact on the space utility, we change location and scale parameter of $Number_{cat}$, scale parameter of $V_{cat}$, clip parameter of $V_{cat}$, width, depth, and height. We have generated a total of five datasets, and their dynamics are progressing. Here we discuss the changes of each dataset compared with the previous one. Dataset one (D1) change the $Number_{cat}$'s location from 1000 to 500. Dataset two (D2) change the $Number_{cat}$'s location from 500 to 100. Dataset three (D3) change the $Number_{cat}$'s

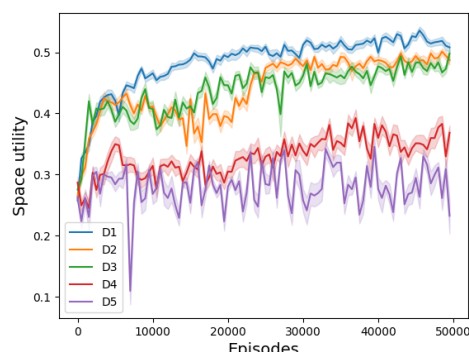

Figure 8: Failure case study result on five datasets with progressing dynamics.

scale from 0.118 to 10. Dataset four (D4) change the $V_{cat}$'s clip from $[1, 4]$ to $[0.5, 8]$. Dataset five (D5) change the clip of width, depth, and height to $[10, 90]$. The result of failure case study is shown in Fig 8, when the increase of data uncertainty is limited, our model **DACC** is stable and can converge to a better space utility. With the substantial increase of data uncertainty (D5), the model fails.

Table 6: Major differences among three simulators: LiftSim, Elevate 8, and our self-developed engine which will be open-sourced.

| Environment | LiftSim | Elevate 8 | Ours |
|---|---|---|---|
| **Meet realistic constraints** | × | ✓ | ✓ |
| **Realistic passenger data** | × | ✓ | ✓ |
| **Support up-down mode** | ✓ | ✓ | ✓ |
| **Support destination mode** | × | ✓ | ✓ |
| **Provide interface for RL** | ✓ | × | ✓ |
| **Open source** | ✓ | × | ✓ |

Table 7: Comparison among different agents in two-way traffic pattern in terms of $T_{aw}$ and $T_{as}$. We set the agent's goal as optimizing the mean waiting time and the average transmitting time.

| Agent | $T_{aw} \downarrow$ | $T_{as} \downarrow$ |
|---|---|---|
| **DACC w/o DAM and rules** | 111.73 | 173.55 |
| **DACC w/o rules** | 65.44 | 128.02 |
| **DACC w/o DAM** | 35.13 | 91.08 |
| **DACC (our full version)** | 32.79 | 89.36 |

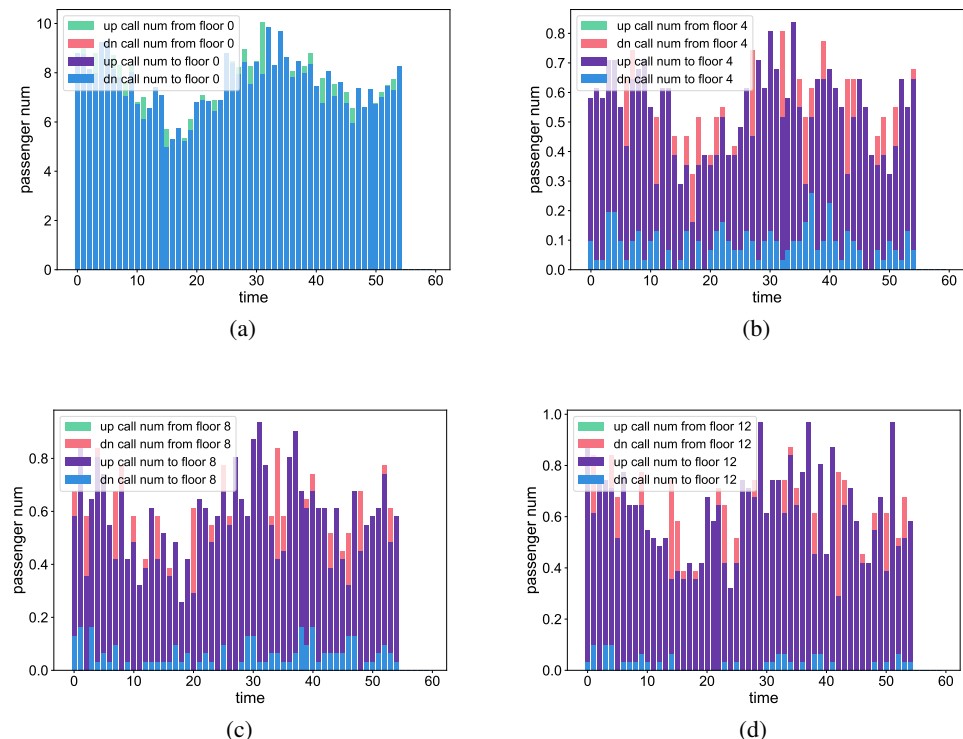

Figure 9: Average number of passengers per minute in one hour in two-way pattern: (a) Floor 0. (b) Floor 4. (c) Floor 8. (d) Floor 12.

## J    OUR ENHANCED SIMULATOR OF THE EGSP

The existing EGSP simulator Liftsim (Wang et al., 2020) fails to meet realistic constraints mentioned in § 4.1 and the commercial EGSP software Elevate8 (Peter) is not open-source for training RL agents. To better reflect the mechanism of a commercial elevator scheduler, we first develop our simulator based on Liftsim, satisfying all the realistic constraints of the real-world EGSP. And we implement both the up-down mode and destination mode for further study. The differences among LiftSim, Elevate 8, and our simulator are concluded in Tab. 6.

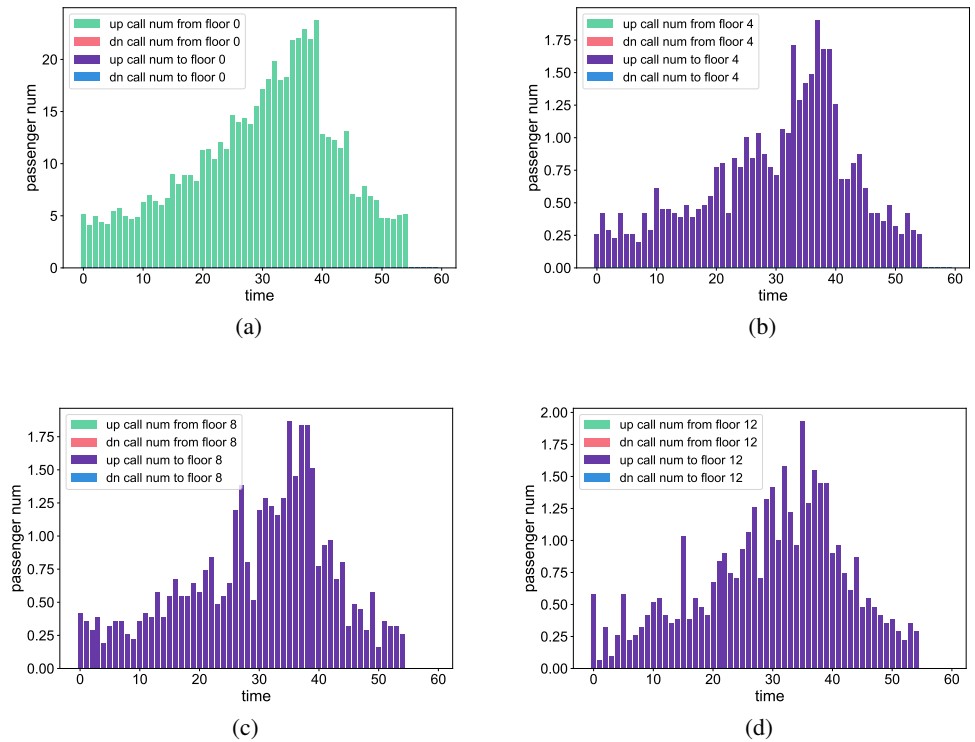

Figure 10: Average number of passengers per minute in one hour in up-peak pattern: (a) Floor 0. (b) Floor 4. (c) Floor 8. (d) Floor 12.

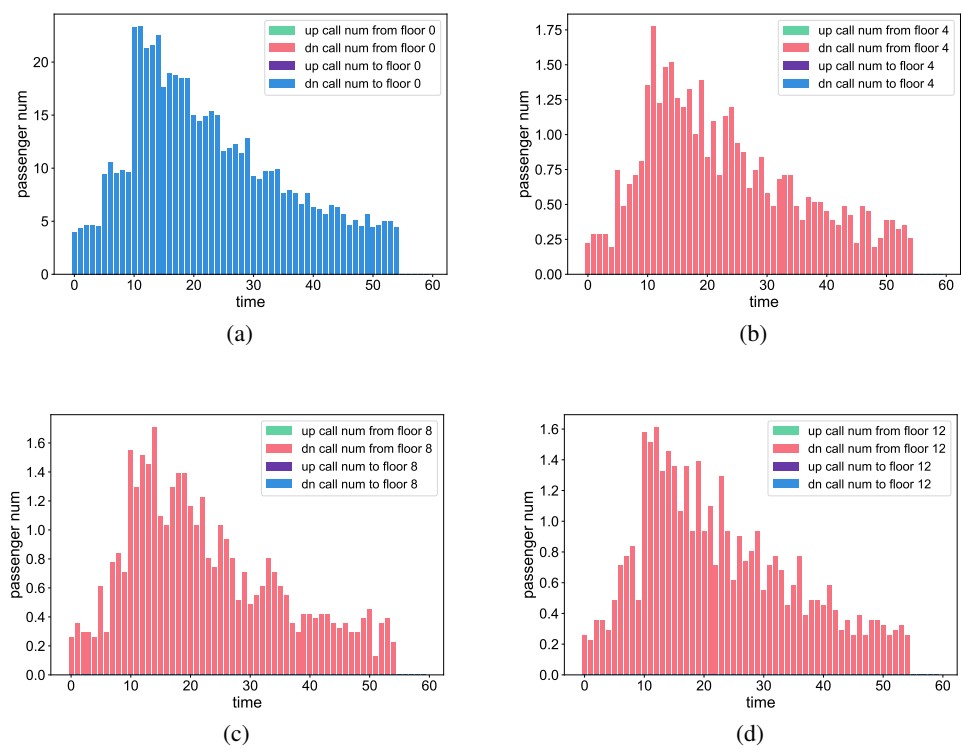

Figure 11: Average number of passengers per minute in one hour in down-peak pattern: (a) Floor 0. (b) Floor 4. (c) Floor 8. (d) Floor 12.

## K    DYNAMIC PASSENGER DATA OF THREE PATTERNS IN OUR SIMULATOR

As for dynamic passenger data, instead of randomly generating passenger data with fixed probability factors, we use real-world passenger data (provided by a world-renowned elevator manufacturer) that conforms to different patterns from a 16-floor building. And we further augment the data with randomness for RL's training. The real-world passenger data of three different flow patterns (two-way pattern, up-peak pattern, and down-peak pattern) are illustrated in Fig. 9, Fig. 10 and Fig. 11, respectively. And for each pattern, we take four floors (floor 0, floor 4, floor 8, and floor12) as examples and show the average number of passengers per minute in one hour for each floor. And for each floor, we show the average number of passengers that request to go up from this floor, go down from this floor, go up to this floor, and go down to this floor. From Fig. 9, we can know that in the two-way pattern, the up call requests are mostly from floor 0, and the down call requests are mostly to floor 0. From Fig. 10, we can know that in the up-peak pattern, up-call requests from floor 0 to other floors form the majority of all hall call requests. From Fig. 11, we can know that in the down-peak pattern, most of the hall call requests are down-call requests from other floors to floor 0. From all three figures, we also know that the total passenger flow change over time. The highest peak time of total passenger flow in an hour is 30 to 35 minutes in the two-way pattern, 35 to 40 minutes in the up-peak pattern, and 12 to 17 minutes in the down-peak pattern.

## L    ABLATION STUDY ON THE EGSP

We take an ablation study to evaluate the advantages of the dynamics aware module and combining rules. In this experiment, we evaluate four agents on a 120-min passenger data flow that follows the two-way traffic mode: the agent without the dynamics aware module (DAM) and rules, the agent without rules, the agent without the dynamics aware module, and the full version of DACC agent—assuming that the objective in this scenario is to reduce the average waiting time and the average transmitting time. From the experimental result shown in Table 7, we can find that both the dynamics aware module and combining rules play essential roles. By combining the advantages of rules and dynamics aware, our framework, DACC, can inherit the controllability and reliability of rules and further optimize the customized objectives for different scenarios.

