# OpenReview forum: "Towards Solving Industrial Sequential Decision-making Tasks under Near-predictable Dynamics via Reinforcement Learning: an Implicit Corrective Value Estimation Approach"
_ICLR.cc/2023/Conference — Submitted to ICLR 2023_

### Official Review · Reviewer_7Ak6 · 2022-10-21

**Confidence:** 3
**Correctness:** 2
**Technical Novelty And Significance:** 2
**Empirical Novelty And Significance:** 3
**Recommendation:** 5

**Clarity, Quality, Novelty And Reproducibility:**

## Quality and Novelty

The authors provide a novel algorithmic framework for understanding reinforcement learning in these input-driven models with empirical results in bin packing and elevator scheduling problems.  However, the model is well studied and the authors provide no new theoretical justification of their algorithm.

## Clarity

The submission is poorly written. The authors should read through the text to fix the mistakes and typos.  For some high level writing comments:
- The "two-stage" model is not described in the introduction
- In the introduction it is mentioned that the demand is highly-predictable and non-stationary, but then later mention that it is near predictable.  This distinction makes sense later on once the model is described - but could be included earlier
- Section 3.1 needs to include a full description of the underlying MDP - i.e. defining $S,A,P$ in this model.  The full transition distribution is never fully described, and the value functions are introduced without ***any*** context
- The discussion on page 4 frequently refers to "bias" but this is never described mathematically.  The remarks that are broken out are also not technical or theoretically justified.

And some minor corrections:
- "especially" twice in the abstract
- "behind GPU computing" unclear in abstract
- "two-stage" MDP in abstract not defined
- "focuses" twice in first paragraph
- "zheng" first paragraph citation
- "decision algorithms" first paragraph
- Last sentence on top of page 3 is unclear
- space after $f_t(d)$ on bottom of page 3
- Paragraph before section 3.2 is unclear
- $i_t$ not defined in 4.1.1
- "set other" in page 7
- $h_t$ space on page 8

**Strength And Weaknesses:**

## Strengths
1. The authors consider an important problem of learning to plan and schedule with empirical results in bin packing and elevator scheduling.

## Weaknesses
1. The writing needs to be improved in order for the main points of the paper to be better described.
2. There are no theoretical justification for the advantages of considering the two-stage model over just the naïve model
3. The related work is insufficient, and the authors do not connect their model to other existing works in the area

**Summary Of The Paper:**

Reinforcement learning can be used in many industrial decision-making problems due to its potential to outperform heuristics.  To avoid issues around scale, current state-of-the-art practical algorithms are simple rule-based strategies which are tuned for improved performance.  In contrast, reinforcement learning is a powerful technique that can be used to learn near optimal policies.  Important to industrial models is the fact that the dynamics are near-predictable (i.e. the decision maker has additional knowledge on the transition distribution).  This is observed in elevator scheduling and bin-packing, where the demands are essentially the only unknown in the problem formulation.  Additional information like this can be used to obtain improved performance.  At a high level, the authors first present these problems as a two-stage MDP, allowing the algorithm to reduce state transition uncertainty.  They then design DACC, a framework for learning input dynamics and making decision with guidance of problem-specific rules to satisfy constraints.  The authors then complement these results with numerical experiments on bin packing and elevator scheduling problems.

More concretely - the authors first state the two-stage MDP model.  In this model the transition can be decomposed into two stages: state-dependent stage and input-dependent stage.  In the first stage, the algorithm picks action $a_t$ based on current arrival $d_t$, leading to a deterministic next state.  In the second stage, the new arrival $d_{t+1}$ is sampled according to a state-independent distribution.

Based on this model, the authors note that the value function decomposes over the exogenous arrival $d_t$ (although this is never proved explicitly, and the authors do not give a general MDP formulation for their model). To complement this reduction they propose a framework for learning the value function estimates (and hence a policy) over this decomposition.

## Questions
- Should it be $F_{t+1}$ and $F_t$ in equation 1?
- Do you believe that learning a good-enough predictor of the demand sequence is reasonable in these settings? See "Protean: VM Allocation Service at Scale" which highlights how demand patterns are highly correlated and impossible to predict.
- What is the justification for "online interaction" in the training versus a historical dataset?

## Related Work
The authors should consider the following papers and how they fit into their model and discussions:
- "OR-Gym: A Reinforcement Learning Library for Operations Research Problems" - considers impact of "action masking" on learning a policy.  The authors seem to suggest that the fact that the problems have constraints make RL in input driven environments more complicated.  However, the constraints that they consider are just the fact that the feasible actions are state-dependent.  Simple action masking (as the authors do in practice) seems to alleviate that issue.
- "Hindsight Learning for MDPs with Exogenous Inputs" - provides a general model for input driven MDPs similar to one studied here - and highlights a decomposition of the value function over the exogenous demand variables (similar to Equation 2)
- "Sample-Efficient Reinforcement Learning in the Presence of Exogenous Information" - Considers RL with extra exogenous information
- "Learning Compact Models for Planning with Exogenous Processes" - similar algorithmic framework for learning a mask in input driven MDP models
- "Markov Decision Processes with Exogenous Variables" - value function decomposition
- "Variance Reduction for Reinforcement Learning in Input-Driven Environments" - This paper was already cited but the authors should include a detailed distinction between their MDP model and the one considered here
-

**Summary Of The Review:**

The authors omit a detailed discussion on the relationship of the proposed research to prior work on exogenous decomposition in MDPs.  Moreover, the writing quality and clarity needs to be improved.  The authors provide no theoretical justification for the benefits of the two-stage decomposition of the process (and similar ideas have been presented in prior work).

---

> ### Author Response · Authors · 2022-11-10
> **Answer to Reviewer 7Ak6**
>
> We thank the reviewer for the provided comments on our work. We aim to address the raised questions/concerns in the following:
>
> >***Q1: Should it be $F_{t+1}$ and $F_t$ in equation 1?***
>
> We have substituted $F_t$ with $f_{t+1:\infty}$ to represent the feature of future distributions and revised Section 3 for a better explanation.
>
> >***Q2: Do you believe that learning a good-enough predictor of the demand sequence is reasonable in these settings? See "Protean: VM Allocation Service at Scale" which highlights how demand patterns are highly correlated and impossible to predict.***
>
> We strongly agree that it is very difficult to directly predict exogenous input, just as the demand patterns in VM Allocation are highly correlated and impossible to predict. Therefore, we do not explicitly predict the next input or future inputs in our framework, instead, we learn the feature of the current/future pattern, which is important for the value estimation of the current state in RL. For instance, the state that all elevators stay on the first floor should have a high value in the up-peak pattern but should have a quite low value in the down-peak pattern. Thus learning the pattern features is critical for accurate value estimation, which further determines whether the RL agent can learn a good policy. Furthermore, in the scheduling cases of the VM allocation service or other general industrial scheduling cases, we believe that it is impossible to accurately predict the demand sequence, but the features of the latent regularity can be utilized by neural networks. And these learned features are quite useful for RL models to better estimate the state value and thus learn good policy in near-predictable industrial settings, which is neglected by most existing works toward industrial RL methods. In addition, we explore the boundary of near-predictability that our tailored model can handle in Appendix I Failure Case Study, in which we test the model's ability to withstand environmental uncertainty.
>
> We do hope our work can raise the attention to such patterns in industry applications which may enlarge the effectiveness and applicability of RL.
>
> >***Q3: What is the justification for "online interaction" in the training versus a historical dataset?***
>
> The prediction problem in RL is to estimate the value of a particular state or state/action pair, given an environment and a policy. The control problem in RL is to find the best policy given an environment.
>
> Solving the control problem using value-based methods involves estimating the value of being in a specific state (i.e., solving the prediction problem) and adjusting the policy to make higher-value choices based on those estimates. This is called generalized policy iteration. The main thing to note here is that the prediction problem is stationary (all long-term expected distributions are the same over time), whilst the control problem adds a non-stationary target for the prediction component (the policy changes, so does the expected return, distribution of states, etc.). Thus, the control problem requires online forgetting behavior from the estimators, an online learning feature.
>
> >***Q4: The authors should consider the following papers and how they fit into their model and discussions.***
>
> Thanks for suggesting these references. We add a discussion of the comparison with the existing RL models with exogenous inputs in Appendix B.
>
> >***Q5: For some high level writing comments:***
> >***1. The "two-stage" model is not described in the introduction***
>
> We revise the introduction, in which we detail the original source of two-stage and our use.
> >***2. In the introduction it is mentioned that the demand is highly-predictable and non-stationary, but then later mention that it is near predictable. This distinction makes sense later on once the model is described - but could be included earlier***
>
> Sorry for giving you a high-predictable awareness of our settings. We check our statement and there is no mention of high-predictable. To eliminate semantic misunderstandings, we reorganize the abstract and introduction and added FAILURE CASE STUDY to the appendix I to visualize our near-predictable.
> >***3. Section 3.1 needs to include a full description of the underlying MDP. The full transition distribution should be described, and the value functions are introduced without any context.***
>
> We have revised Section 3.1 to now include a full description of the underlying MDP and a full transition distribution. And we introduce the value function in Section 3.2.
> >***4. The discussion on page 4 frequently refers to "bias" but this is never described mathematically. The remarks that are broken out are also not technical or theoretically justified.***
>
> We have added the relevant mathematical proofs in Appendix C and D.
> >***Q6: And some minor corrections: ...***
>
> Thank you for your patient pointing out and kind reminder. We have revised them accordingly in our newly uploaded version.

---

> > ### Comment · Reviewer_7Ak6 · 2022-11-19
> > **Response**
> >
> > Thanks for responding to the comments and submitting an updated revision - it was easy to track all of the changes with the use of colors as well.  The presentation in Section 3 is much better at highlighting the two-stage MDP formulation, especially in describing it in terms of the typical MDP primitives.  I also appreciate the modification of the DGP for the problems in the appendix, since it highlights how the demand change is near predictable in these settings.
> >
> > My concerns are still around comparison to existing algorithms in the field (especially that presented by Mao 2019, which seems directly applicable to these problems).  Did you include any variance-reduction framework from the model when training the A2C/DACC algorithms? Potentially those would be the easiest to modify to help complete the story when comparing the approach to the related literature.

---

> > > ### Author Response · Authors · 2022-12-06
> > > **Answer to Reviewer 7Ak6**
> > >
> > > Thank you for spending additional time reviewing our (significantly) updated revision.
> > >
> > >
> > > First, we use the [open-source](https://github.com/hongzimao/input_driven_rl_example) of Mao's paper and compare it with ours in the two scenarios (EGP and BPP). For EGP, we evaluate the two methods with 10 random input sequences (each with about 900 passengers following the two-way traffic pattern) and use the average waiting time (AWT) as an evaluation metric. For BPP, in order to run the model within acceptable time and memory, we simplified the scenario, changing the pallet size from 120\*100\*100 to 12\*10\*10, and reducing the item size proportionally. We evaluate both methods on 100 random sequences and use space utility (SU) and average item num (AIN) as criteria. The results are shown as follows:
> > > |                    | EGP (AWT$\downarrow$) | BPP (SU$\uparrow$)|  BPP(AIN$\uparrow$) |
> > > | --------------- | ----------------------------------- | ---------------------------|------------------ |
> > > | DACC            | **46.99 $\pm$ 2.09** | **69.87% $\pm$ 6.49%**  | **34.10 $\pm$ 5.73**|              |
> > > | Mao 2019 | 48.41 $\pm$ 4.47 | 21.58% $\pm$ 7.71%     | 12.04 $\pm$ 4.44  |
> > >
> > >
> > > From the table, we can see that in our setting, our method (DACC) achieves better performance with lower std, especially in the BPP case. We conjecture (also from their source code) that Mao's method is more suited to the case when the input keeps relatively constant over time with noise modeling. While our approach is more designed for the case when the sequences show certain dynamics (but not constant) which is nearly predictable.
> > >
> > >
> > > We are also keeping seeking other variance reduction frameworks for comparison.

---

### Official Review · Reviewer_CzjD · 2022-10-22

**Confidence:** 3
**Correctness:** 3
**Technical Novelty And Significance:** 3
**Empirical Novelty And Significance:** 3
**Recommendation:** 6

**Clarity, Quality, Novelty And Reproducibility:**

Clarity: The structure of the paper and the mathematics are fine. The flow of arguments leaves something to be desired, e.g., why exactly what the authors are doing is necessary. The paper seems to be combining a lot of stuff, but it is not really clear how the combination works (and, in the end, the performance is only a little better than A2C?).

Quality: The work is justified by intuition and experiments. I would prefer to see more thorough experiments that analyze the pieces of DACC (see weaknesses).

Novelty: The source of innovation in this paper comes from the two-stage MDP, a new interpretation for the non-stationary near predictable environment. The DACC uses two critic networks to learn environment constraints and task-related value functions respectively.

Reproducibility: The algorithms are described clearly. It is unclear if the authors will release code.

**Strength And Weaknesses:**

Strengths:
1. The problem setting and methods are well-defined.  The two example tasks are representative for the application environment targeted in this paper.
2.  The two-stage  MDP  proposed is technically correct as its two stages learn the constraints of the non-stationary environment and the task-specific or time-dependent values respectively.
3.  The experiments compare the proposed methods  (DACC)  with  4  heuristic methods and  2  other learning-based methods as well as two learning-based models for ablation studies in two example tasks.

Weaknesses:
1.  The DACC models use a marginal latent variable F(t) that is dependent on the recent experiences to account for the ”near predictable” feature of the problem.  However, the agent might only exploit recent regularity instead of the long-term regularity that is motivated by the authors (e.g., in elevator scheduling)
2.  Due to the recent experience-dependent F(t), how does the model respond to change of input pattern (e.g., is there a delay, can things go wrong during the transition?)
3.  The two critic networks in DACC use each other’s estimation of value function to update themselves. Does this create issues? What is the impact of the KL divergence term—does changing the weight of the KL divergence term affect the outcome?
4. The safety mask is not present in A2C and "DACC w/o rules", the two models in the paper's ablation study. I think the DACC w/o rules has more parameters to train and hence should be more powerful than A2C. But the experiment data shows that A2C has better performance. So if the extra critic in DACC w/o rules doesn't necessarily increase the performance, then the safety mask might be more important than the extra critic. Can A2C be tested with a safety mask?

**Summary Of The Paper:**

This  paper  proposes  a  new  Markov  Decision  Process  (MDP)  definition  to account  for  the  non-stationary near-predictable  tasks  that  are  common  in  industrial  application.   These tasks require the agents to learn the constraints and rules of environment and the strategy to get high reward in the dynamic environment.  The author propose a two-stage MDP in which the agent learns constraints of the environment and the near-predictable reward separately in two steps.  A RL model DACC is designed to fit in the two-stage MDP setting.  With two critic networks, one of which learns the constrained transition of environment states and the other learns the transition of state reward that are regular to some extent.  The experiments compare DACC to other methods on two representative applications.  Two models (A2C and DACC w/o rules) are included in experiments as an ablation study trying to support the superiority of DACC’s dual critic design. These models outperform others.

**Summary Of The Review:**

The framework is interesting and the model is novel.  However, the experiment design failed to prove that the superiority of DACC comes from its two-stage design but not safety mask that can significantly reduce the action space. The dual critic set up raises convergence concerns.

Post author revisions:
The clarity of the paper is improved. A more inclusive ablation study supports the superiority of the model DACC and the two-stage MDP. The detailed training algorithm helps to explain how the critics are updated during training. The ablation also clarifies that A2C w/o rules performs worse than DACC w/o rules, while A2C with rules performs better than DACC w/o rules.

---

> ### Author Response · Authors · 2022-11-10
> **Answer to Reviewer CzjD**
>
> We thank the reviewer for the feedback and the suggestions for improving our experiments. We now answer to the raised questions and discuss the suggestions regarding baselines and experiments.
>
> >***Q1: The DACC models use a marginal latent variable $F(t)$ that is dependent on recent experiences to account for the "near predictable" charactersitics of our problem setting. However, the agent might only exploit recent regularity instead of the long-term regularity that is motivated by the authors (e.g., in elevator scheduling)***
>
> We agree with your considerations, and before our main experiments, we also explore how much of the interval is enough for long-sighted scheduling. And we found that as it takes about two minutes for an elevator to traverse sixteen floors, thirty minutes (the length of recent experiences we use in our paper) of recent experiences contain sufficient long-term information for the agent to make decisions. Secondly, because the Dynamics-aware module is independent of the decision module, it can be pre-trained and migrated, not just for a few minutes of specific data, but for learning more general pattern features. This strength of our framework is verified by the migration experiment in our new version (in Appendix H).
>
> >***Q2: Due to the recent experience-dependent F(t), how does the model respond to change of input pattern (e.g., is there a delay, can things go wrong during the transition?)***
>
> To clarify what the input pattern looks like, we visualize the passenger flow of three patterns (up-peak, down peak and two-way pattern) in Appendix K. As there exist obvious regularities for each pattern, the major duty of the Dynamics-aware module is to map recent experients to a hidden space, which is then fed to critic networks for value estimation of the current state. What's more, in real-world industrial scenarios, we pay more attention to how the model handles the impact of data variability in the same pattern rather than the transition between different patterns. In reality, there is usually a relatively long transition period between different patterns. For example, in elevator scheduling, the up-peak is generally from 7 a.m. to 10 a.m., the lunch peak is from 11 a.m. to 1 p.m., and the down-peak is from 5 p.m. to 7 p.m., and the interval between different patterns is relatively long.
>
> >***Q3: Does two critic networks in DACC updating themselves with each other's estimation of value function create issues? What is the impact of the KL divergence term and its weight?***
>
> We add formal derivation in Section 3.2 and Appendix D. The relationship between the two estimated values is shown in Eq.4, which is also the justification for adding the KL divergence loss to the overall loss function. Along the training process, the KL loss drives $z^i_{t+1}$ to keep a distribution close to $z^s_t$, which indirectly enforces the relationship between $V^i(s_t, f_{t:\infty})$ and $V^s(s_t, d_t, f_{t+1:\infty})$. As this process is somehow a joint training of the two critic networks, the weight of KL loss can not be set too large, otherwise, it will introduce extra variance because $V^i(s_t, f_{t:\infty})$ is overfitted to $V^s(s_t, d_t, f_{t+1:\infty})$ in a training step, instead of getting close to its expectation gradually.
>
> >***Q4: Why does A2C perform better than "DACC w/o rules"? Can A2C be tested with a safety mask?***
>
> Thank you for pointing this out. In our first version, the compared A2C model is actually equipped with rules, and we lack the ablated model A2C without rules. We are sorry that we did not clarify that, and this led to a misunderstanding. In our latest version, we specify the ablated model A2C with rules and A2C without rules. As illustrated in Fig. 4 in Section 4.1.2, DACC can converge to stable performance earlier with higher reward than A2C. And after adding rules, the gap between DACC and A2C is further enlarged, which means that our model can better utilize the rule guidance. It is also worth noting that in this ablation study, the neural network structures of the DACC model and the A2C model are the same. Thus the comparison is fair.
>
> >***Q5: It is unclear if the authors will release the code.***
>
> We upload an unclean version of our source codes, including our framework in both cases and our enhanced elevator simulator. We will release a clean version in GitHub later for further research.

---

### Official Review · Reviewer_EhZ3 · 2022-10-24

**Confidence:** 2
**Correctness:** 2
**Technical Novelty And Significance:** 3
**Empirical Novelty And Significance:** 3
**Recommendation:** 5

**Clarity, Quality, Novelty And Reproducibility:**

- Clarity. See the previous section on weaknesses.
- Quality. The claims made in Section 3 are reasonable and should be correct. The experiment results cannot be verified but seem plausible.
- Novelty. The paper focuses on a novel setting and provided new insights into industrial-scale decision making problems
- Reproducibility. The paper cannot be reproduced at its current stage.

**Strength And Weaknesses:**

Strength:
- The topic covered by the paper is intriguing and significant for enabling wider adoption of RL algorithms.
- The illustrations provided in the paper help better illustrate the core ideas

Weaknesses:
- The paper can be hard to understand at times. For instance, in Section 3.1, it took several reads to realize what the authors mean by "two-stage" framework. I also personally believe that writing $S_t = (S_t^p, d, F_t)$ could better represent this fact. It is also relatively unclear to me what **exactly** $F_t$ represents: in the paper immediately above eqn (1) it is mentioned that $F_t$ relates to the marginal distribution of future boxes and is related to a series of distributions. Are we saying that $F_t$ is some function that could approximate the future distributions? Approximate in what sense? Are we saying that $F_t$ is a model that is able to predict $f_t(d)$ using information obtained up to step $t$?
- While the figures contain error bars but standard errors are not reported in the tables. The number of runs used to obtain the error bars is not reported.
- The authors use a self-developed simulator for the elevator task. It is hard to assess what the simulator does by simply reading Appendix B without seeing the underlying source code.

**Summary Of The Paper:**

The paper aims to solve industrial sequential decision making tasks. The paper makes two observations about the transition dynamics for industrial scale decision making tasks and proposes a two-stage approach that leverages the underlying structure of industrial problems. Using the observation the paper proposes a bi-critic framework (DACC) which leverages this underlying structure. Finally, empirical results are provided for the proposed method.

**Summary Of The Review:**

The topic studied in the paper is novel and interesting. The paper would significantly benefit from improved presentation and more instructions on how to reproduce the results. Further details on the experiments are also be greatly appreciated.

---

> ### Author Response · Authors · 2022-11-10
> **Answer to Reviewer EhZ3**
>
> We thank the reviewer for the effort put into reading our paper and the detailed comments. In the following, we aim to address all the concerns.
>
> >***Q1: In Section 3.1, it took several reads to realize what the authors mean by the "two-stage" framework.***
>
> We have rewritten Section 3.1, updated Fig. 1, and added a comparison with the single-stage formulation in Fig. 6 to better explain the proposed two-stage MDP and our DACC framework.
>
> >***Q2: What does $F_t$ exactly represent? Are we saying that $F_t$ is some function that could approximate the future distributions? Approximate in what sense? Are we saying that $F_t$ is a model that is able to predict $f_t(d)$ using information obtained up to step $t$?***
>
> Thanks for your feedback. In our new version, we change $F_t$ to $f_{t+1:\infty}$ for a better understanding. As the paper explains, $F_t$ (now denoted as $f_{t+1:\infty}$) is a feature vector representing the future distributions. We want to emphasize that we do not explicitly predict $f_t(d)$. (The reason is that it is difficult to give a precise estimate even in the near-predictable setting. And it makes little sense to utilize the predicted $f_t(d)$ with significant errors.) Instead, we use the learned features $F_t$ for the critic networks to estimate the value of the current state. This is important because, in industrial scenarios, the value of the current state is highly related to the characteristics of future data. For instance, in the EGSP, the value of the state where all elevators are waiting on the first floor should be very large in the up-peak pattern while quite small in the down-peak pattern because passengers mostly appear in the first floor in the up-peak pattern but hardly appear in the first floor in down-peak. So we argue that the dynamic aware module plays a vital role in extracting features of future distributions and promoting value estimation of the critic networks.
>
> >***Q3: While the figures contain error bars, standard errors are not reported in the tables. The number of runs used to obtain the error bars is not reported.***
>
> Thank you for spotting this cursoriness. In the BPP, we evaluate each baseline and our model with 50 runs. In EGSP, we generate 20 random sequences of passengers that follow specific traffic patterns to evaluate each method. We are now re-running the experiments to get the error bar of all compared methods and will update the tables in a near future version.
>
> >***Q4: The authors use a self-developed simulator for the elevator task. Can you show the underlying source code of the simulator?***
>
> We add the detailed data generation process of BPP and EGSP in Appendix  G and K. And we visualize the characteristics of three passenger patterns in EGSP to help explain what exact regularities $F_t$ focuses on. And we have uploaded the source codes of our enhanced elevator simulator. We will release a cleaner version of our project on GitHub.

---

### Official Review · Reviewer_WYQB · 2022-10-25

**Confidence:** 3
**Correctness:** 3
**Technical Novelty And Significance:** 3
**Empirical Novelty And Significance:** 3
**Recommendation:** 5

**Clarity, Quality, Novelty And Reproducibility:**

### Clarity

Sec 3.1 is important for the audience to understand the contributions of this work. However, it’s currently hard to follow. Please consider revising this part. It would be helpful to directly compare the two stage framework with the standard one-stage framework in Figure 1 and provide some formal analysis. The statement that existing RL methods suffer from high estimation bias is too vague.

### Quality

There are some grammar issues and typos. Please carefully proofread the draft.

### Novelty

The proposed method is somewhat novel.



**Strength And Weaknesses:**

### Strength

The proposed two-stage MDP framework is novel and empirical results show that it perform well in non-stationary industrial benchmark tasks.

### Weaknesses

The clarity needs improvement. See detailed comments below.

**Summary Of The Paper:**

For certain industrial sequential decision problems, this paper proposes to decompose the conventional MDP transition into two steps, state-dependent stage and input-dependent stage, and learn separate value functions. The state-dependent stage focuses on security or safe constraints and the input-dependent stage is for capturing the non-stationarity of the environment. The authors design a dynamic-aware and constraints-confined reinforcement learning based on this two-stage MDP formulation. On bin packing problems and elevator group scheduling problems, the proposed method demonstrate superior performance compared to rule-based algorithms and learning based algorithms.

**Summary Of The Review:**

This paper studies the important problem of applying RL in industrial applications and the proposed method demonstrates strong empirical performance. I could be willing to accept this paper if the clarity could be improved.

---

> ### Author Response · Authors · 2022-11-10
> **Answer to Reviewer WYQB**
>
> We thank the reviewer for the positive feedback and suggestions, allowing us to improve the paper further. In the following, we answer the raised concerns.
> >***Q1: Sec 3.1 is important for the audience to understand the contributions of this work. However, it’s currently hard to follow. Please consider revising this part.***
>
> Thank you for your suggestion. We have rewritten Section 3.1 and updated Fig. 1 and Fig. 2 to explain better the proposed two-stage MDP and DACC framework in our newly updated pdf.
>
> >***Q2: It would be helpful to directly compare the two-stage framework with the standard one-stage framework in Figure 1 and provide some formal analysis.***
>
> Thank you for your helpful advice. Due to space limitations, we show the standard single-stage framework in Fig. 6 in Appendix C. We further provide a formal derivation of DACC's advantage in reducing the variance of value estimation in Appendix C and D.
>
> >***Q3: The statement that existing RL methods suffer from high estimation bias is too vague.***
>
> Thank you for pointing this out. In our latest version, we revise high estimation bias to high estimation variance for existing RL methods, which is caused by the dynamics uncertainty and the restricted action space in industrial scenarios. And we refer to the proof in [VRRL] and claim that our framework does not introduce extra bias while reducing variance.
>
> [VRRL] Mao, Hongzi, et al. "Variance Reduction for Reinforcement Learning in Input-Driven Environments." International Conference on Learning Representations. 2018.

---

> > ### Author Response · Authors · 2022-12-06
> > **Answer to Reviewer WYQB**
> >
> > We rewrote Section 3 and clarified all important concepts separately, which has been acknowledged.  Could you please take a look at our new version and tell us if there is anything unclear? Looking forward to your feedback.

---

### Author Response · Authors · 2022-11-10
**General response to all reviewers/AC: the revision of our paper**

We thank all the reviewers for their insightful comments. We have revised our paper accordingly and improved our presentation. We highlight the main changes to the paper in blue in the updated pdf. The main changes are summarized as follows:
1. We revise Section 3 and update Fig. 1 and Fig. 2 to improve the presentation of our proposed two-stage MDP and our DACC framework. We provide a mathematical derivation of our framework's advantage in reducing value estimation variance in Appendix C and Appendix D.
2. We revise the related work Section and extend it in Appendix A and Appendix B to highlight our paper’s novelty and merits compared to existing works.
3. We add more details of the training algorithms for better reproducibility in Appendix E. We upload an internal version of our source code, including our framework in the two cases and our enhanced elevator simulator (which we believe very useful for the community). The final publicly released version may be further cleaned.
4. We include a couple of new experiments: i) ablated models of A2C with rules, A2C without rules, and DACC without rules (Sec 4.1.2) for showing the superiority of DACC over A2C; ii) generalization experiments (Appendix H) indicating that DACC can perform well even in the untrained dataset; iii) failure case study (Appendix I) to explore the extent of dynamics uncertainty that our model can handle in practice.
5. We depict the detailed data generation process of BPP and EGSP in Appendix G and K, respectively. We also visualize the characteristics of three passenger patterns in EGSP (Fig. 9, Fig. 10, and Fig. 11) to clarify that the passenger flow change over time even in the same pattern, which are the near-predictable regularities we focus on in this paper.

---

### Author Response · Authors · 2022-11-18
**Dear Reviewers:**

Approaching the pdf updating DDL, is there anything needing added?

---

### Decision · Program_Chairs · 2023-01-20

**Decision:**

Reject

**Justification For Why Not Higher Score:**

The proposed formulation is essentially a classical technique for representation / modeling in RL, but unfortunately the authors do not seem to know this.

**Justification For Why Not Lower Score:**

N/A

**Metareview: Summary, Strengths And Weaknesses:**

This paper proposes an RL framework motivated by many industrial applications, where we can isolate the stochasticity arising from state-independent distribution. The experiments were done on two simulated industrial tasks, bin packing and elevator dispatching.

The manuscript has improved considerably after the initial review phase. Unfortunately, it seems that the paper is essentially re-discovering a classical technique in RL. The proposed formulation seems essentially the same as the afterstate representation (Sutton & Barto the RL book) or the post-decision state representation (Powell 2007). How the proposed framework is different or related to this classical technique should be adequately addressed through literature review.


**Summary Of Ac-Reviewer Meeting:**

The reviewers highly appreciated that the paper is tackling an important problem of scheduling / planning core to the OR community. The  empirical results were also a positive point about the paper.

The lack of literature review was a negative point, and especially the lack of comparison to the classical RL representation technique known as afterstate or post-decision state was the major reason behind the rejecting the paper.